# Dynamic and Efficient Gray-Box Hyperparameter Optimization for Deep Learning

## Abstract

Gray-box hyperparameter optimization techniques have recently emerged as a promising direction for tuning Deep Learning methods. However, the multi-budget search mechanisms of existing prior works can suffer from the poor correlation among the performances of hyperparameter configurations at different budgets. As a remedy, we introduce DYHPO, *a method that learns to dynamically decide which configuration to try next, and for what budget*. Our technique is a modification to the classical Bayesian optimization for a gray-box setup. Concretely, we propose a new surrogate for Gaussian Processes that embeds the learning curve dynamics and a new acquisition function that incorporates multi-budget information. We demonstrate the significant superiority of DYHPO against state-of-the-art hyperparameter optimization baselines through large-scale experiments comprising 50 datasets (Tabular, Image, NLP) and diverse neural networks (MLP, CNN/NAS, RNN).

## 1 Introduction

Hyperparameter Optimization (HPO) is arguably an acute open challenge for Deep Learning (DL), especially considering the crucial impact HPO has on achieving state-of-the-art empirical results. Unfortunately, HPO for DL is a relatively under-explored field and most DL researchers still optimize their hyperparameters via obscure trial-and-error practices. On the other hand, traditional Bayesian Optimization HPO methods (Snoek et al., 2012; Bergstra et al., 2011) are not directly applicable to deep networks, due to the infeasibility of evaluating a large number of hyperparameter configurations. In order to scale HPO for DL, three main directions of research have been recently explored. *(i) Online HPO* methods search for hyperparameters during the optimization process via meta-level controllers (Chen et al., 2017; Parker-Holder et al., 2020), however, this online adaptation can not accommodate all hyperparameters (e.g. related to architectural changes). *(ii) Gradient-based HPO* techniques, on the other hand, compute the derivative of the validation loss w.r.t. hyperparameters by reversing the training update steps (Maclaurin et al., 2015; Franceschi et al., 2017; Lorraine et al., 2020), however, the reversion is not directly applicable to all cases (e.g. dropout rate). The last direction, *(iii) Gray-box HPO* techniques discard sub-optimal configurations after evaluating them on lower budgets (e.g. after few epochs) (Li et al., 2017; Falkner et al., 2018).

In contrast to the online and gradient-based alternatives, gray-box approaches can be deployed in an off-the-shelf manner to all types of hyperparameters and architectures. The gray-box concept is based on the intuition that a poorly-performing hyperparameter configuration can be identified and terminated by inspecting the validation loss of the first few epochs, instead of waiting for the full convergence. The most prominent gray-box algorithm is Hyperband (Li et al., 2017), which runs random configurations at different budgets (e.g. different number of epochs) and successively halves these configurations by keeping only the top performers. Follow-up works, such as BOHB (Falkner et al., 2018) or DEHB (Awad et al., 2021), replace the random sampling of Hyperband with more intelligent sampling based on Bayesian optimization or differentiable evolution.

Despite their great practical potential, existing gray-box methods suffer from a major issue. The low-budget (few epochs) performances are not always a good indicator for the full-budget (full convergence) performances. For example, a properly regularized network converges slower in the first few epochs, however, typically performs better than a non-regularized variant after the full convergence. In other words, there is a poor rank correlation of the configurations' performances

at different budgets. As a remedy, **we introduce DYHPO, a gray-box method that *dynamically* decides how many configurations to try and how much budget to spend on each configuration.**

DYHPO is a Bayesian Optimization (BO) approach based on Gaussian Processes (GP), that extends and adapts BO to the multi-budget (a.k.a. multi-fidelity) case. In this perspective, we propose a deep kernel GP that captures the learning dynamics (learning curve until the evaluated budget). As a result, we train a kernel capable of capturing the similarity of a pair of hyperparameter configurations, even if the pair's configurations are evaluated at different budgets (technically, different learning curve lengths). Furthermore, we extend Expected Improvement (Jones et al., 1998) to the multi-budget case, by introducing a new mechanism for the incumbent configuration of a budget.

The joint effect of modeling a GP kernel across budgets together with a dedicated acquisition function leads to DYHPO achieving a statistically significant empirical gain against state-of-the-art gray-box baselines (Falkner et al., 2018; Awad et al., 2021), including prior work on multi-budget GPs (Kandasamy et al., 2017; Metz et al., 2020). We demonstrate the performance of DYHPO in three diverse popular types of deep learning architectures (MLP, CNN/NAS, RNN) and 50 datasets of three diverse modalities (tabular, image, natural language processing). We believe our method is a step forward towards making HPO for DL practical and feasible. Overall, our contributions can be summarized as follows:

- We introduce a novel Bayesian surrogate model that is based on a Gaussian Process with a deep kernel. This surrogate model predicts the validation score of a machine learning model based on the hyperparameter configuration and the learning curve.

- We derive a simple yet robust way to combine this surrogate model with Bayesian optimization, reusing most of the existing components currently used in traditional Bayesian optimization methods.

- Finally, we demonstrate the efficiency of our method for hyperparameter optimization and neural architecture search tasks compared to the current state-of-the-art methods. As an overarching goal, we believe our method is an important step towards scaling HPO for DL.

## 2 MOTIVATION

As mentioned earlier, a major source of sub-optimality for the current gray-box techniques (Li et al., 2017; Falkner et al., 2018; Awad et al., 2021) is the poor rank correlation of performances at low and high budgets, which is endemic to the successive halving mechanism. In essence, prior methods mistakenly discard good hyper-parameter configurations by myopically relying only on the early performance after few epochs. In contrast, our method fixes the endemic poor correlation of successive halving, by learning to decide which configuration to continue optimizing for one more time-budget step (e.g. one more epoch). In that perspective, our strategy resembles Freeze-Thaw (Swersky et al., 2014), however, differs in fitting deep kernel GP surrogates with learning curves and a special multi-budget acquisition. DYHPO never discards a configuration as all options remain latent until the acquisition selects the most promising one and allocates one more budget unit of optimization.

We illustrate the differences between our strategy and successive halving with the experiment of Figure 1, where we showcase the HPO progress of three different methods on the "Helena" dataset from the LCBench benchmark (Zimmer et al., 2021). Random search is an example of a black-box approach that trains each candidate until completion without considering the intermediate scores. Hyperband (Li et al., 2017) is a gray-box approach that *statically* pre-allocates the budget for a set of candidates (Hyperband bracket) according to a predefined policy. Finally, DYHPO *dynamically* adapts the allocation of budgets for configurations after every HPO step.

The plots in the top row of Figure 1 show the learning curves of multiple neural networks trained with different hyperparameter configurations. The darker the color of a learning curve, the later the model corresponding to the learning curve was trained during the optimization process. In an optimal scenario, there should be no learning curve of darker color trained for a very long time if there is already a learning curve of lighter color with higher accuracy. Since black-box functions do not consider intermediate responses, this trend is not observed for random search. This is also not the case for gray-box methods such as Hyperband, because it is possible that no configuration selected for a Hyperband bracket is outperforming the current best configuration. Since at least one

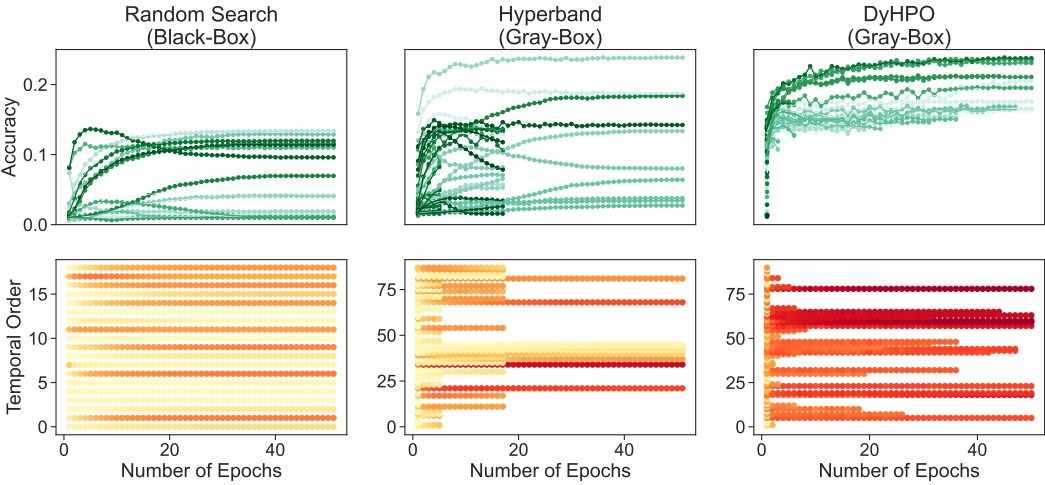

Figure 1: Top row: Learning curves observed during the search. The darker the learning curve, the later it was evaluated during the search. Bottom row: y-axis shows a sequence of learning curve evaluations (bottom to top). The color indicates accuracy. The darker red the higher the accuracy.

of the bracket candidates must be fully trained according to the predefined policy, we observe this suboptimal behavior in Figure 1. This motivates our idea of dynamically deciding when we want to continue training a configuration based on the intermediate response. Clearly, DYHPO invests only a small budget into configurations that show little promise as indicated by the intermediate scores.

## 3 RELATED WORK ON GRAY-BOX HPO

**Multi-Fidelity Bayesian Optimization.** Bayesian optimization is a black-box function optimization framework that has been successfully applied in optimizing hyperparameter and neural architectures alike (Snoek et al., 2012; Kandasamy et al., 2018; Bergstra et al., 2011). To further improve Bayesian optimization, several works propose low-fidelity data approximations of hyperparameter configurations. Low-fidelity approximations of hyperparameter configurations can be obtained by training on a subset of the data (Swersky et al., 2013; Klein et al., 2017) or terminating training early (Swersky et al., 2014). Several methods extend Bayesian optimization to multi-fidelity data by engineering new kernels suited for this problem (Swersky et al., 2013; 2014; Poloczek et al., 2017). Kandasamy et al. (2016) extends GP-UCB (Srinivas et al., 2010) to the multi-fidelity setting by learning one Gaussian Process (GP) with a standard kernel for each fidelity. Their later work improves upon this method by learning one GP for all fidelities that enables the use of continuous fidelities (Kandasamy et al., 2017). The work by Takeno et al. (2020) follows a similar idea but proposes to use an acquisition function based on information gain instead of UCB. While most of the works rely on GPs to model the surrogate function, Li et al. (2020) use a Bayesian neural network that models the complex relationship between fidelities with stacked neural networks, one for each fidelity.

**Multi-Fidelity Bandits.** Hyperband (Li et al., 2017) is a bandits-based multi-fidelity method for hyperparameter optimization, that due to its simplicity and strong performance, enjoys great popularity. The algorithm selects hyperparameter configurations at random and uses successive halving (Jamieson & Talwalkar, 2016) with different settings to early-stop less promising training runs. Several improvements have been proposed to Hyperband with the aim to replace the random sampling of hyperparameter configurations with a more guided approach (Bertrand et al., 2017; Wang et al., 2018; Wistuba, 2017). BOHB (Falkner et al., 2018) uses TPE (Bergstra et al., 2011) and builds a surrogate model for every fidelity adhering to a fixed-fidelity selection scheme. DEHB (Awad et al., 2021) samples candidates using differential evolution which handles discrete and large hyperparameter search spaces better than BOHB.

**Deep Kernel Learning with Bayesian Optimization.** We are among the first to use deep kernel learning with Bayesian optimization and to the best of our knowledge the first to use it for multi-fidelity Bayesian optimization. Rai et al. (2016) consider the use of a deep kernel instead of a manually designed kernel in the context of standard Bayesian optimization, but, limit their experimentation to synthetic data and do not consider its use for hyperparameter optimization. Perrone et al. (2018); Wistuba & Grabocka (2021) use a pre-trained deep kernel to warm start Bayesian optimization with meta-data from previous optimizations. These approaches are multi-task or transfer learning methods that require the availability of meta-data from related tasks.

In contrast to prior work, we propose the first deep kernel GP for multi-fidelity HPO that is able to capture the learning dynamics across fidelities/budgets, combined with an acquisition function that is tailored for the gray-box setup. Furthermore, our work represents an important step towards scaling HPO for Deep Learning (DL), by demonstrating a statistically significant reduction in terms of HPO time on a series of DL network architectures and a large set of diverse datasets.

## 4 Preliminaries

**Black-Box Optimization.** As mentioned earlier, the problem of optimizing hyperparameters can be modeled as a black-box optimization problem. The objective is to maximize the response function $f : \mathcal{X} \to \mathbb{R}$ that returns the validation score for training a machine learning model with a hyperparameter configuration $\mathbf{x} \in \mathcal{X}$. In practice, this observation is noisy such that we observe in fact $y_i = f(\mathbf{x}_i) + \varepsilon$ where $\varepsilon \sim \mathcal{N}(0, \sigma_n^2)$.

**Gray-Box Optimization.** Since many machine learning algorithms allow to measure at various fidelities, a relaxation of the black-box to the gray-box optimization problem is in many cases logical and allows for significantly faster optimization. The gray-box setting allows us to query configurations with a budget smaller than the maximum budget $B$. Thus, we can query from the response function $f : \mathcal{X} \times \mathbb{N} \to \mathbb{R}$ where $f_{i,j} = f(\mathbf{x}_i, j)$ is the response after spending a budget of $j$ on configuration $\mathbf{x}_i$. As before, these observations are noisy and we observe $y_{i,j} = f(\mathbf{x}_i, j) + \varepsilon_j$ where $\varepsilon_j \sim \mathcal{N}(0, \sigma_{j,n}^2)$. Please note, we assume that the budget required to query $f_{i,j+b}$ after querying $f_{i,j}$ is only $b$. While this is not necessarily the case for all problems, the models we consider are learned incrementally. Furthermore, we are able to make use of the learning curve $\mathbf{Y}_{i,j-1} = (y_{i,1}, \ldots, y_{i,j-1})$ when predicting $f_{i,j}$.

**Bayesian Optimization (BO).** BO is a general framework for solving black-box optimization problems and is very popular for optimizing hyperparameters. It has two ingredients, i.e. a surrogate model and an acquisition function. The surrogate is a probabilistic model which approximates the black-box function using the available information of function evaluations. The acquisition function returns the expected utility for a configuration given the surrogate model's prediction. The BO framework executes the following steps sequentially. First, the surrogate model is trained on the available information about the black-box function, denoted as the history of evaluations $\mathcal{D}$. Then, the configuration with the highest expected utility $\mathbf{x}_i$ is evaluated and $y_i$ is observed. The tuple $(\mathbf{x}_i, y_i)$ is added to $\mathcal{D}$ and the process is repeated until the HPO budget is exhausted. A common choice for the surrogate model are Gaussian Processes (Rasmussen & Williams, 2006), a popular choice for the acquisition function is expected improvement (Jones et al., 1998).

**Gaussian Processes (GP).** GPs are probabilistic machine learning models. Given a training data set $\mathcal{D} = \{(\mathbf{x}_i, y_i) | i = 1, \ldots, n\}$ with $n$ data points, the Gaussian Process assumption is that $y_i$ is a random variable and the joint distribution of all $y_i$ is assumed to be multivariate Gaussian distributed as:

$$\mathbf{y} \sim \mathcal{N}\left(m\left(\mathbf{X}\right), k\left(\mathbf{X}, \mathbf{X}\right)\right) \tag{1}$$

Furthermore, $\mathbf{f}_*$ for test instances $\mathbf{x}_*$ are jointly Gaussian with $\mathbf{y}$ as:

$$\begin{bmatrix} \mathbf{y} \\ \mathbf{f}_* \end{bmatrix} \sim \mathcal{N}\left(m\left(\mathbf{X}, \mathbf{x}_*\right), \begin{pmatrix} \mathbf{K}_n & \mathbf{K}_* \\ \mathbf{K}_*^T & \mathbf{K}_{**} \end{pmatrix}\right) . \tag{2}$$

The mean function $m$ is often set to $\mathbf{0}$ and its covariance function $k$ depends on parameters $\boldsymbol{\theta}$. For notational convenience, we use:

$$\mathbf{K}_n = k\left(\mathbf{X}, \mathbf{X} | \boldsymbol{\theta}\right) + \sigma_n^2 \mathbf{I}, \ \mathbf{K}_* = k\left(\mathbf{X}, \mathbf{X}_* | \boldsymbol{\theta}\right), \ \mathbf{K}_{**} = k\left(\mathbf{X}_*, \mathbf{X}_* | \boldsymbol{\theta}\right) \tag{3}$$

to define the kernel matrix. We can derive the posterior predictive distribution with mean and co-variance as follows:

$$\mathbb{E}\left[\mathbf{f}_*|\mathbf{X}, \mathbf{y}, \mathbf{X}_*\right] = \mathbf{K}_*^T \mathbf{K}_n^{-1} \mathbf{y}, \ \mathrm{cov}\left[\mathbf{f}_*|\mathbf{X}, \mathbf{X}_*\right] = \mathbf{K}_{**} - \mathbf{K}_*^T \mathbf{K}_n^{-1} \mathbf{K}_* \tag{4}$$

Often, the kernel function is manually engineered, one popular example is the squared exponential kernel. However, in this work, we make use of the idea of deep kernel learning (Wilson et al., 2016). The idea is to model the kernel as a neural network $\varphi$ and learn the best kernel transformation as:

$$k_{\mathrm{deep}}(\mathbf{x}, \mathbf{x}'|\boldsymbol{\theta}, \mathbf{w}) = k(\varphi(\mathbf{x}, \mathbf{w}), \varphi(\mathbf{x}', \mathbf{w})|\boldsymbol{\theta}). \tag{5}$$

As we will see later, this allows us to use convolutional operations as part of our kernel.

## 5 DYNAMIC MULTI-FIDELITY HYPERPARAMETER OPTIMIZATION

In this section, we will describe DYHPO, our proposed method for hyperparameter optimization in the gray-box setting. At first, we will describe the surrogate model which is a Gaussian Process with a deep convolutional kernel. Then, we describe a variation of the popular expected improvement acquisition function (Jones et al., 1998), modified to consider multiple fidelities, and conclude with the final algorithm.

### 5.1 MULTI-FIDELITY SURROGATE WITH DEEP CONVOLUTIONAL KERNEL

We propose to use a Gaussian Process surrogate model that infers the value of $f_{i,j}$ based on the hyperparameter configuration $\mathbf{x}_i$, the budget $j$ as well as the past learning curve $\mathbf{Y}_{i,j-1}$. For this purpose, we use a deep kernel which is defined as

$$k_{\mathrm{deep}}((\mathbf{x}_i, \mathbf{Y}_{i,j-1}, j), (\mathbf{x}_{i'}, \mathbf{Y}_{i',j'-1}, j')) = k(\varphi(\mathbf{x}_i, \mathbf{Y}_{i,j-1}, j), \varphi(\mathbf{x}_{i'}, \mathbf{Y}_{i',j'-1}, j')) \tag{6}$$

We use a squared exponential kernel for $k$ and the neural network $\varphi$ is composed of linear and convolutional layers as shown in Figure 2. We normalize the budget $j$ to a range between $0$ and $1$ by dividing it by the maximum budget $B$. Afterward, it is concatenated with the hyperparameter configuration $\mathbf{x}_i$ and fed to a linear layer.

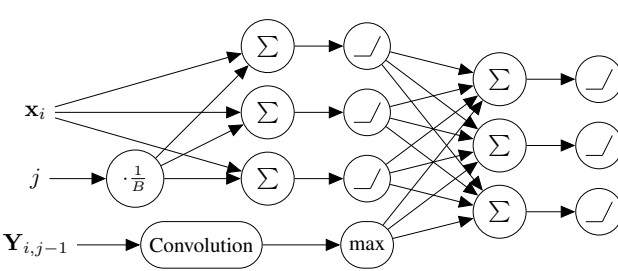

The learning curve $\mathbf{Y}_{i,j-1}$ is transformed by a one-dimensional convolution followed by a global max pooling layer. Finally, both representations are fed to another linear layer. Its output will be the input to the kernel function $k$. Both, the kernel $k$ and the neural network $\varphi$ consist of trainable parameters $\boldsymbol{\theta}$ and $\mathbf{w}$, respectively. We find their optimal values by computing the maximum likelihood estimates as:

Figure 2: The feature extractor $\varphi$ used in our deep kernel.

$$\hat{\boldsymbol{\theta}}, \hat{\mathbf{w}} = \underset{\boldsymbol{\theta}, \mathbf{w}}{\arg\max} \, p(\mathbf{y}|\mathbf{X}, \mathbf{Y}, \boldsymbol{\theta}, \mathbf{w}) = \underset{\boldsymbol{\theta}, \mathbf{w}}{\arg\max} \int p(\mathbf{y}|\mathbf{X}, \mathbf{Y}, \boldsymbol{\theta}, \mathbf{w}) p(\mathbf{f}|\mathbf{X}, \mathbf{Y}, \boldsymbol{\theta}, \mathbf{w}) d\mathbf{f}$$
$$\propto \underset{\boldsymbol{\theta}, \mathbf{w}}{\arg\min} \, \mathbf{y}^{\mathrm{T}} \mathbf{K}_n^{-1} \mathbf{y} + \log|\mathbf{K}_n| \tag{7}$$

In order to solve this optimization problem, we use gradient descent and Adam (Kingma & Ba, 2015) with a learning rate of $0.1$. Given the maximum likelihood estimates, we can approximate the predictive posterior as formalized in Equation 8, and ultimately compute the mean and covariance of this Gaussian using Equation 4.

$$p\left(f_{i,j}|\mathbf{x}_i, \mathbf{Y}_{i,j-1}, j, \mathcal{D}\right) \approx p\left(f_{i,j}|\mathbf{x}_i, \mathbf{Y}_{i,j-1}, j, \mathcal{D}, \hat{\boldsymbol{\theta}}, \hat{\mathbf{w}}\right) \tag{8}$$

## 5.2 Multi-Fidelity Expected Improvement

Expected improvement (Jones et al., 1998) is a commonly used acquisition function for the black-box setting and is defined as:

$$\text{EI}(\mathbf{x}|\mathcal{D}) = \mathbb{E}\left[\max\left\{f(\mathbf{x}) - y^{\text{max}}, 0\right\}\right] , \tag{9}$$

where $y^{\text{max}}$ is the largest observed value of $f$. We propose a multi-fidelity version of it as:

$$\text{EI}_{\text{MF}}(\mathbf{x}, j|\mathcal{D}) = \mathbb{E}\left[\max\left\{f(\mathbf{x}, j) - y_j^{\text{max}}, 0\right\}\right] , \tag{10}$$

where:

$$y_j^{\text{max}} = \begin{cases} \max\left\{y \mid ((\mathbf{x}, \cdot, j), y) \in \mathcal{D}\right\} & \exists \mathbf{x} \in \mathcal{X} : ((\mathbf{x}, \cdot, j), y) \in \mathcal{D} \\ \max\left\{y \mid (\cdot, y) \in \mathcal{D}\right\} & \text{otherwise} \end{cases} \tag{11}$$

Simply put, $y_j^{\text{max}}$ is the largest observed value of $f$ for a budget of $j$ if it exists already, otherwise, it is the largest observed value for any budget. If there is only one possible budget, the multi-fidelity expected improvement is identical to expected improvement.

## 5.3 The DyHPO Algorithm

---
**Algorithm 1:** DyHPO Algorithm

---
1: **while** not converged **do**
2:     $\mathbf{x}_i \leftarrow \arg\max_{\mathbf{x} \in \mathcal{X}} \text{EI}_{\text{MF}}(\mathbf{x}, j)$ (Section 5.2)
3:     Observe $y_{i,j}$.
4:     $\mathcal{D} \leftarrow \mathcal{D} \cup \left\{((\mathbf{x}_i, \mathbf{Y}_{i,j-1}, j), y_{i,j})\right\}$
5:     Update the surrogate model on $\mathcal{D}$. (Section 5.1)
6: **return** $\mathbf{x}_i$ with largest $y_{i,j}$.

---

The DyHPO algorithm looks very similar to many black-box Bayesian optimization algorithms as shown in Algorithm 1. The big difference is that at each step we dynamically decide which candidate configuration to train *for a small additional budget*. Possible candidates are previously unconsidered configurations as well as configurations that did not reach the maximum budget. In Line 2, the most promising candidate is chosen using the acquisition function introduced in Section 5.2 and the surrogate model's predictions. It is important to highlight that we do not maximize the acquisition function along the budget dimensionality. Instead, we set $j$ such that it is by exactly one higher than the budget used to evaluate $\mathbf{x}_i$ before. If $\mathbf{x}_i$ has not been evaluated for any budget yet, $j$ is set to 1. This ensures that we explore configurations by slowly increasing the budget. After having selected the candidate and the corresponding budget, the function $f$ is evaluated and we observe $y_{i,j}$ (Line 3). This additional data point is added to $\mathcal{D}$ in Line 4. Then in Line 5, the surrogate model is updated according to the training scheme described in Section 5.1.

## 6 Experiments

We evaluate DyHPO in three different settings on hyperparameter optimization for tabular, text, and image classification against several competitor methods. These include Hyperband (Li et al., 2017), BOHB (Falkner et al., 2018), DEHB (Awad et al., 2021), and Dragonfly (Metz et al., 2020). We use Dragonfly's multi-fidelity optimizer (Kandasamy et al., 2017). For a sanity check, we also compare against random search (Bergstra & Bengio, 2012). We use the publicly available implementations whenever available and implemented Hyperband and random search ourselves. We report the mean of ten repetitions and report two common metrics such as the regret and the average rank. The regret refers to the absolute difference between the score of the solution found by an optimizer compared to the best possible score. If we report the regret as an aggregate result over multiple datasets, we report the mean over all regrets. The average rank is a metric we use to aggregate results over different datasets. For each dataset, the best performing method obtains a rank of 1. Ties are broken by using the average rank, e.g., if the methods have scores 0.9, 0.8, 0.8, 0.7, the ranks are 1, 2.5, 2.5, and 4. For both metrics, smaller is better.

### 6.1 Feedforward Neural Networks

In our first experiment, we evaluate the various methods on how well they optimize neural networks for tabular datasets. For this purpose, we use the LCBench learning curve benchmark (Zimmer et al.,

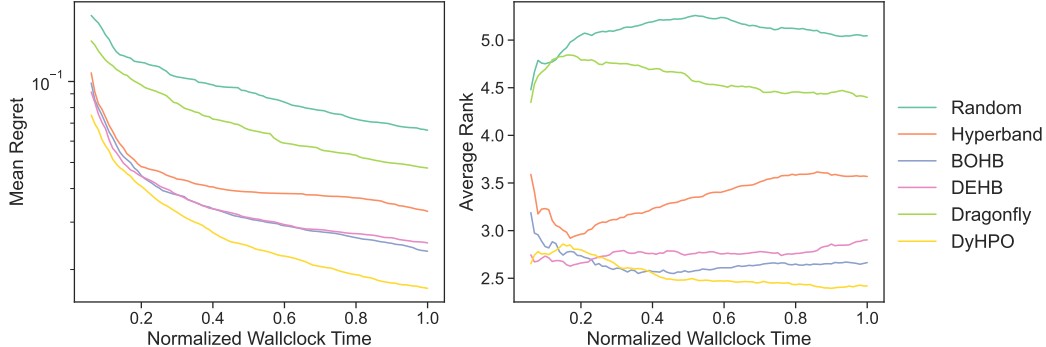

Figure 3: LCBench: Aggregated results over 35 different datasets. The normalized wallclock time represents the actual runtime divided by the total wallclock time of DYHPO including the overhead of fitting the deep GP. DYHPO achieves the best performance among all methods for both metrics.

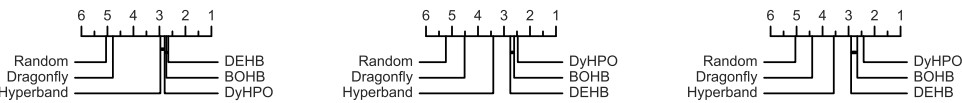

Figure 4: Critical difference diagram for LCBench for results corresponding to the time DYHPO took to complete 200, 600 and 1,000 epochs. DYHPO's improvement is statistically significant.

2021). This benchmark contains learning curves for 35 different datasets where 2,000 neural networks per dataset are trained for 50 epochs with Auto-PyTorch. The objective is to optimize seven different hyperparameters of funnel-shaped neural networks, i.e., batch size, learning rate, momentum, weight decay, dropout, number of layers, and maximum number of units per layer. We show the aggregated results in Figure 3. Here we aggregate the normalized wallclock time by dividing the actual wallclock time of baselines by the total wallclock time of our method DYHPO including the overhead incurred by fitting the deep GP. In that manner, we can aggregate wallclock times across datasets. However, all the results for each individual dataset are reported in the Appendix (Figure 12 and 13). Furthermore, we use a critical difference diagram to report the pairwise statistical difference between all methods (Figure 4). Horizontal lines indicate groups of methods that are not significantly different from each other. As suggested in the work of Demsar (2006), we use the Friedman test to reject the null hypothesis followed by a pairwise posthoc analysis based on the Wilcoxon signed-rank test ($\alpha = 0.05$). While all gray-box methods have a very similar performance in the very beginning, DYHPO quickly outperforms its competitors with respect to both evaluation metrics with statistical significance. On this task, both BOHB and DEHB are significantly better than Hyperband but there is no clear winner among these two methods.

## 6.2 RECURRENT NEURAL NETWORKS

We continue with evaluating all methods on NLP tasks using search spaces provided in TaskSet (Metz et al., 2020). The objective is to optimize eight hyperparameters for a set of different recurrent neural networks (RNN) that differ in embedding size, RNN cell, and other architectural features. The set of hyperparameters consists of optimizer-specific hyperparameters, such as the learning rate, the exponential decay rate of the first and second momentum of Adam, $\beta_1$ and $\beta_2$, and Adam's constant for numerical stability $\varepsilon$. Furthermore, there are two hyperparameters controlling linear and exponential learning rate decays, as well as L1 and L2 regularization terms. As before, we provide the aggregated results in the main paper (Figure 5) and provide detailed results in the Appendix (Figure 11). We show the critical difference diagram in Figure 6. As before, all gray-box methods have a very similar performance in the very beginning, but, DYHPO quickly outperforms its competitors with respect to both evaluation metrics. The difference is significant in the first interval measured, in fact, DYHPO is providing the best results across all different tasks for the majority

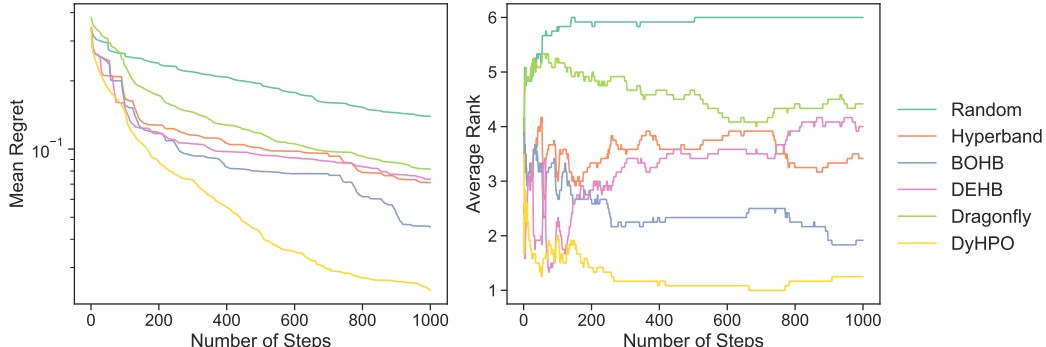

Figure 5: TaskSet: Aggregated results over 12 different NLP tasks. Again, DYHPO shows the best performance among all methods for both evaluation metrics.

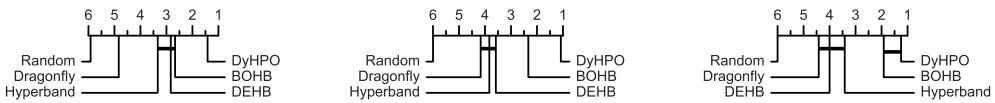

Figure 6: Critical difference diagram for TaskSet for results after 200, 600 and 1,000 epochs, respectively. DYHPO's improvement is statistically significant.

of the elapsed time. Given enough time, BOHB is able to catch up but DYHPO's improvement is still statistically significant.

### 6.3 CONVOLUTIONAL NEURAL NETWORKS

Neural Architecture Search (Zoph & Le, 2017) (NAS) raised a lot of interest in the deep learning community in the last few years. Since it can be reformulated to a hyperparameter optimization problem, we can apply our method as well as many other standard hyperparameter optimization methods to it. We refrain from comparing against specialized NAS methods, since, this is out-of-scope for our work but refer the interested reader to the comparison of Dong & Yang (2020). To evaluate our methods in this scenario, we use NAS-Bench-201 (Dong & Yang, 2020). This benchmark has precomputed about 15,600 architectures trained for 200 epochs for the image classification datasets CIFAR-10, CIFAR-100, and ImageNet. The objective is to select for each of the six operations within the cell of the macro architecture one of five different operations. All other hy-

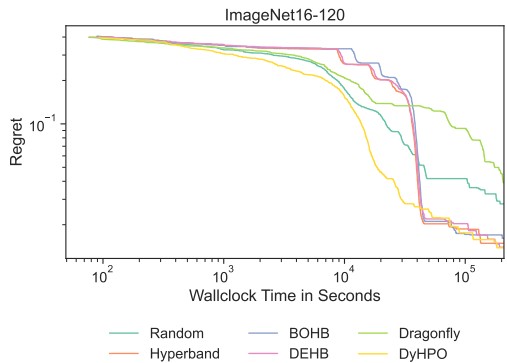

Figure 8: DYHPO quickly finds well-performing configurations. Given enough time, most methods find equally good architectures.

perparameters such as learning rate and batch size are kept fix. We report the results in Figure 8, the remaining results can be found in the Appendix (Figure 10). Initially, DYHPO provides better results, but, given enough time, most methods perform no longer significantly different. We see also no difference between Hyperband and BOHB or DEHB. A reason for this observation will be discussed in the next section.

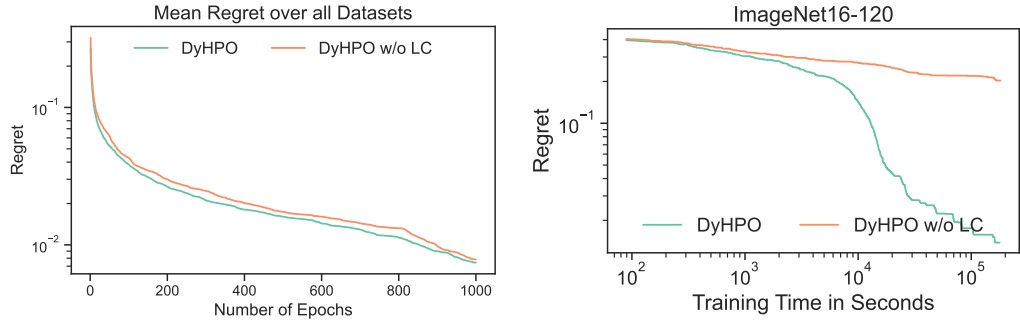

Figure 7: Left: Aggregated results for LCBench. Right: Results on ImageNet from NAS-Bench 201. Using the learning curve gives only little advantage on average for the LCBench problems. However, not using it significantly worsens it for the NAS-Bench 201 tasks.

## 6.4 ABLATION STUDY

One of the main differences between DYHPO and similar methods such as the work by Kandasamy et al. (2017), is that the learning curve is an input to the kernel function. For this reason, we investigate the impact of this design choice. We consider a variation of DYHPO, DYHPO w/o LC, which is identical to its counterpart, and the only difference is that the learning curve is not part of the input. We report the results in Figure 7 and refer to the Appendix for additional results (Figure 9). One of the striking observations is that the learning curve has only a small impact for LCBench in contrast to NAS-Bench-201 where it is significant. Our hypothesis is that for LCBench the hyperparameter representation is a valuable feature such that the additional use of the learning curve (which in fact is already implicitly considered by the Gaussian Process) is not required. For NAS-Bench 201, however, the architecture representation is poor and does not allow for good predictions on its own. This hypothesis aligns well with our previous experiments, where BOHB did better on LCBench compared to Hyperband whereas we have not seen any difference on NAS-Bench 201. As a reminder, BOHB is a variation of Hyperband that considers the hyperparameter representation to sample candidates.

We conclude, that the additional use of an explicit learning curve representation might not lead to a strong improvement in every scenario, however, it also does not seem to deteriorate it. Furthermore, there are cases where the explicit consideration leads to significantly better results.

## 7 LIMITATIONS OF OUR PAPER

Although DYHPO shows a convincing and statistically significant reduction of the HPO time on diverse Deep Learning (DL) experiments, we cautiously characterized our method only as a "step towards" scaling HPO for DL. The reason for our restrain is the lack of tabular benchmarks for HPO on very large deep learning models, such as Transformers-based architectures (Devlin et al., 2019). We hope our promising results will motivate the DL community, that own compute power, to invest in creating pre-computed tabular HPO benchmarks for very deep models, where HPO researchers can demonstrate the empirical performance of their proposed methods.

## 8 CONCLUSIONS

In this work, we present DYHPO, a new Bayesian optimization (BO) algorithm for the gray-box setting. We introduced a new surrogate model for BO that uses a learnable deep kernel and takes the learning curve as an explicit input. Furthermore, we motivated a variation of expected improvement for the multi-fidelity setting. Finally, we compared our approach on diverse benchmarks on a total of 50 different tasks against the current state-of-the-art methods on gray-box hyperparameter optimization (HPO). Our method shows significant gains and has the potential to become the de facto standard for HPO in Deep Learning.

ACKNOWLEDGMENTS

Double-blind.

ETHICS STATEMENT

In our work, we use only publicly available data with no privacy concerns. Furthermore, our algorithm reduces the overall time for fitting deep networks, therefore, saving computational resources and yielding a positive impact on the environment. Moreover, our method can help smaller research organizations with limited access to resources to be competitive in the deep learning domain, which reduces the investment costs on hardware. Although our method significantly reduces the time taken for optimizing a machine learning algorithm that achieves peak performance, we warn against running our method for an extended time only to achieve marginal gains in performance, unless it is mission-critical. Last but not least, in order to save energy, we invite the community to create sparse benchmarks with surrogates, instead of dense tabular ones.

REPRODUCIBILITY STATEMENT

We attempt to facilitate reproduction of our results with the following measures:

- We use only publicly available datasets and provide a detailed description of preprocessing (Section A.2) and the datasets themselves (Section A.1).
- All our baselines are publicly available or trivial to implement. We provide all required details in Section A.5.
- We clearly describe our method in Section 5 and provide additional details in Section A.4.
- Finally, we plan to make the source code of our method publicly available.

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

# A  EXPERIMENTAL SETUP

## A.1  BENCHMARKS

**LCBench.**  LCBench[1] is a feedforward neural network benchmark on tabular data which consists of 2000 configuration settings for each of the 35 datasets. The configurations were evaluated during HPO runs with AutoPyTorch. LCBench features a search space of 7 numerical hyperparameters, where every hyperparameter configuration is trained for 50 epochs.

**TaskSet.**  TaskSet[2] is a benchmark that features over 1162 diverse tasks from different domains and includes 5 search spaces. In this work, we focus on NLP tasks and we use the Adam8p search space with 8 continuous hyperparameters. We refer to Figure 11 for the exact task names considered in our experiments. The learning curves provided in TaskSet report scores after every 200 iterations. We refer to those as "steps" in Figure 5.

**NAS-Bench-201.**  NAS-Bench-201[3] is a benchmark consisting of 15625 hyperparameter configurations representing different architectures on the CIFAR-10, CIFAR-100 and ImageNet datasets. NAS-Bench-201 features a search space of 6 categorical hyperparameters and each architecture is trained for 200 epochs.

## A.2  PREPROCESSING

In the following, we describe the preprocessing applied to the hyperparameter representation. For LCBench, we apply a log-transform to batch size, learning rate, and weight decay. For TaskSet, we apply it on the learning rate, L1 and L2 regularization terms, linear and exponential decay of the learning rate. All continuous hyperparameters are scaled to the range between 0 and 1 using sklearn's MinMaxScaler. If not mentioned otherwise, we use one-hot encoding for the categorical hyperparameters. As detailed in the following, some baselines have a specific way of dealing with them. In that case, we use the method recommended by the authors.

## A.3  FRAMEWORK

The framework contains the evaluated hyperparameters and their corresponding validation curves. The list of candidate hyperparameters is passed to the baseline-specific interface, which in turn, optimizes and queries the framework for the hyperparameter configuration that maximizes utility. Our framework in turn responds with the validation curve and the cost of the evaluation. In case a hyperparameter configuration has been evaluated previously up to a budget $b$ and a baseline requires the response for budget $b + 1$, the cost is calculated accordingly only for the extra budget requested.

## A.4  IMPLEMENTATION DETAILS

We implement the Deep Kernel Gaussian Process using GPyTorch 1.5 (Gardner et al., 2018). We use an RBF kernel and the dense layers of the transformation function $\varphi$ (Figure 2) have 128 and 256 units. We used a convolutional layer with kernel size three and four filters. All parameters of the Deep Kernel are estimated by maximizing the marginal likelihood. We achieve this by using gradient ascent and Adam (Kingma & Ba, 2015) with a learning rate of 0.1 and batch size of 64. We stop training as soon as the training likelihood is not improving for 10 epochs in a row or we completed 1,000 epochs. For every new data point, we start training the GP with its old parameters to reduce the required effort for training.

---

[1]`https://github.com/automl/LCBench`
[2]`https://github.com/google-research/google-research/tree/master/task_set`
[3]`https://github.com/D-X-Y/NAS-Bench-201`

### A.5 BASELINES

**Random Search & Hyperband.** Random search and Hyperband sample hyperparameter configurations at random and therefore the preprocessing is irrelevant. We have implemented both from scratch and use the recommended hyperparameters for Hyperband, i.e. $\eta = 3$.

**BOHB.** For our experiments with BOHB, we use version 0.7.4 of the officially-released code[4].

**DEHB.** For our experiments with DEHB, we use the official public implementation[5]. We developed an interface that communicates between our framework and DEHB. In addition to the initial preprocessing common for all methods, we encode categorical hyperparameters with a numerical value in the interval [0, 1]. For a categorical hyperparameter $\mathbf{x}_i$, we take $N$ equal-sized intervals, where $N_i$ represents the number of unique categorical values for hyperparameter $\mathbf{x}_i$ and we assign the value for a categorical value $n \in N_i$ to the middle of the interval $[i, i+1]$ as suggested by the authors. For configuring the DEHB algorithm we used the default values from the library.

**Dragonfly.** We use the publicly available code of Dragonfly[6]. No special treatment of categorical hyperparameters is required since Dragonfly has its own way to deal with them. We use version 0.1.6 with default settings.

## B    ADDITIONAL PLOTS

In this section, we provide additional plots for the performance comparison between all methods for the individual datasets in our benchmarks. In Figure 12 and 13 we show the performance comparison for all the datasets from LCBench regarding regret over the number of epochs. As can be seen, DYHPO manages to outperform the other competitors in the majority of the datasets, and in the datasets that it does not, it is always close to the top-performing method and the difference between methods is marginal.

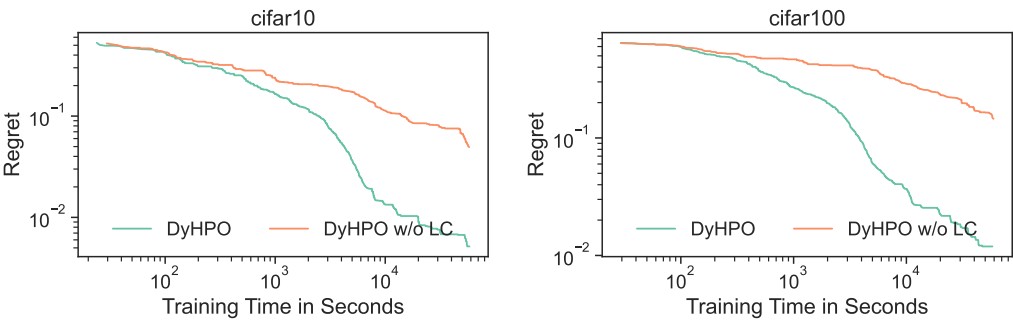

Figure 9: The learning curve as an explicit input is very important for each task of NAS-Bench 201.

Additionally, in Figure 10 we show the performance comparison over time of every method for the CIFAR-10 and CIFAR-100 datasets in the NAS-Bench-201 benchmark. As can be seen, DYHPO converges faster and has a better performance compared to the other methods over the majority of the time, however, towards the end although it is the optimal method or close to the optimal method, the difference in regret is not significant anymore.

Furthermore, Figure 11 shows the performance comparison for the datasets chosen from TaskSet over the number of steps. Looking at the results, DYHPO is outperforming all methods convincingly on the majority of datasets by converging faster and with significant differences in the regret evaluation metric.

---

[4]https://github.com/automl/HpBandSter
[5]https://github.com/automl/DEHB/
[6]https://github.com/dragonfly/dragonfly

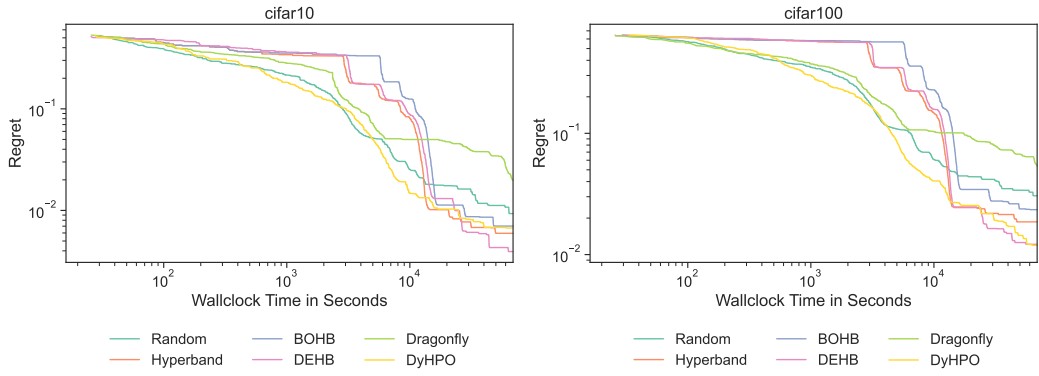

Figure 10: NAS-Bench-201 Regret Results.

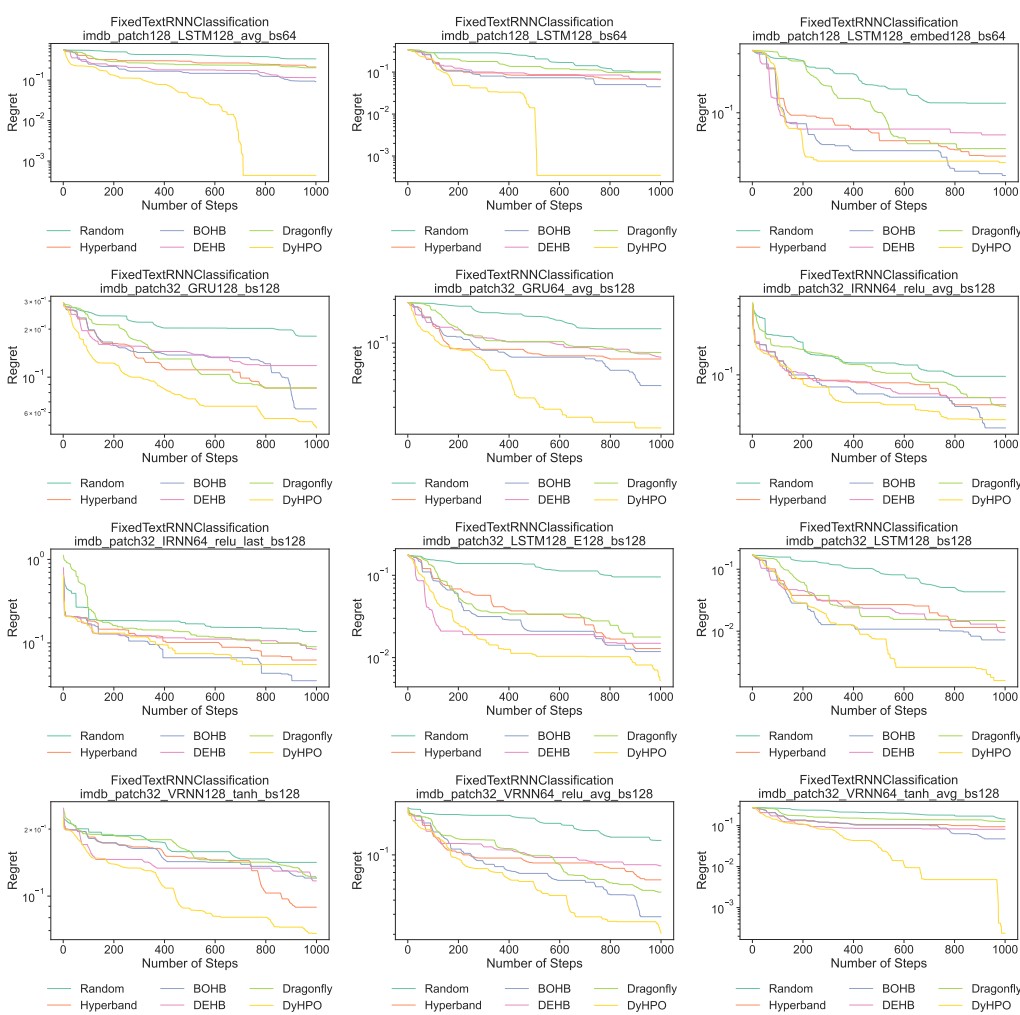

Figure 11: Detailed results on a per dataset level for TaskSet.

Lastly, in Figure 9 we ablate the learning curve input in our kernel, to see the effect it has on performance for the CIFAR-10 and CIFAR-100 datasets. The results indicate that the learning curve plays an important role in achieving better results by allowing faster convergence and a better anytime performance.

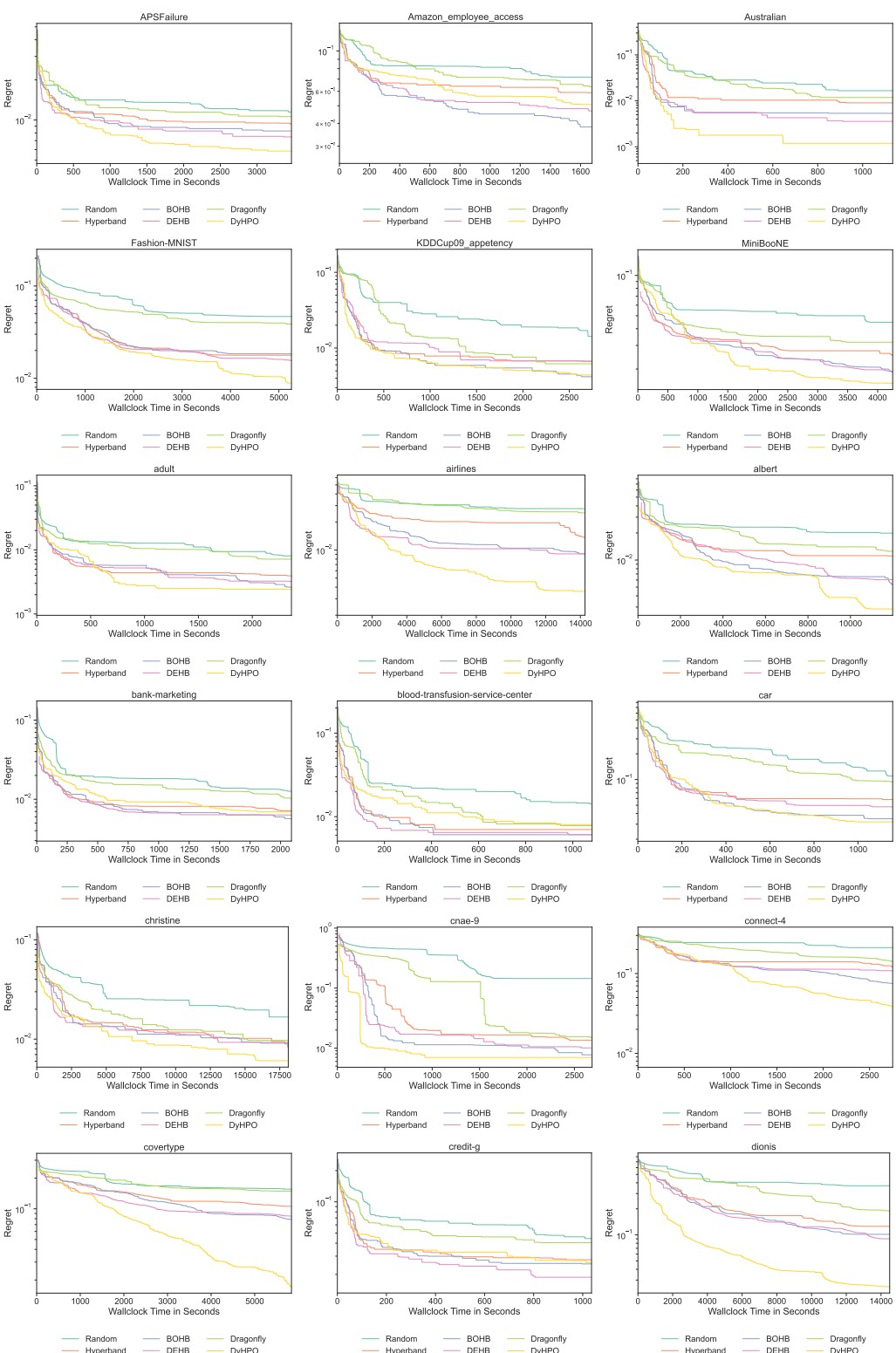

Figure 12: Detailed results on a per dataset level for LCBench.

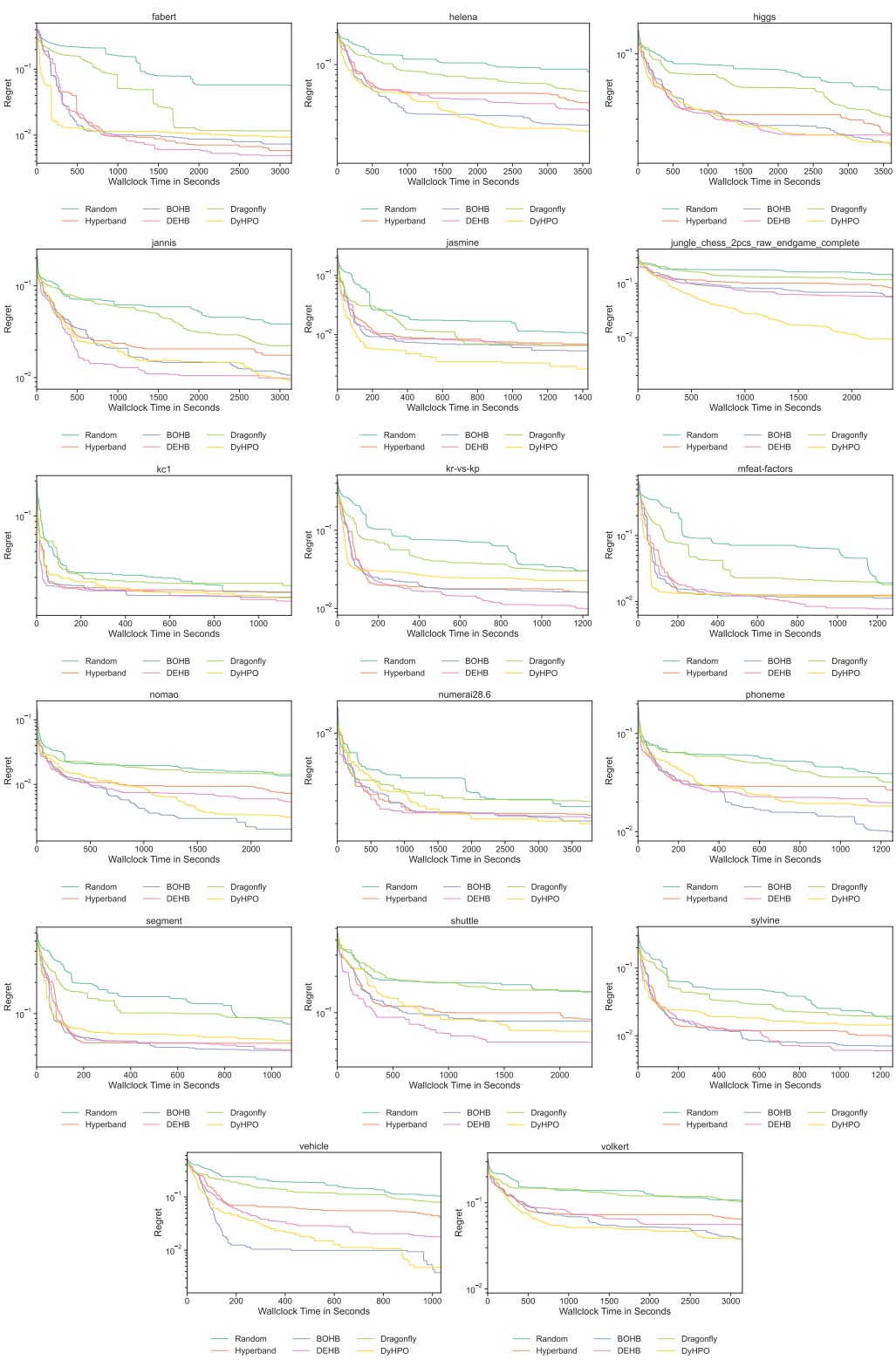

Figure 13: Detailed results on a per dataset level for LCBench (cont.)

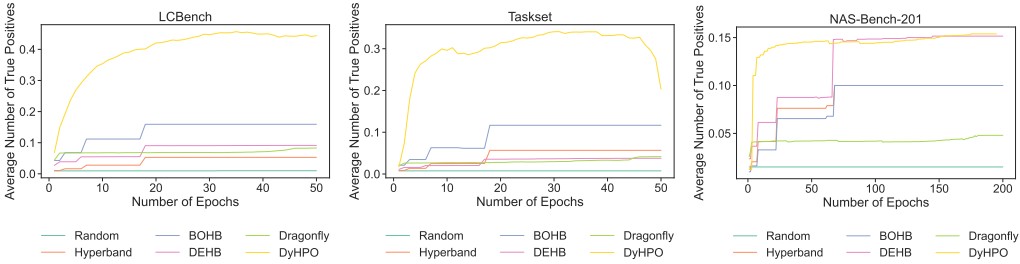

Figure 14: DYHPO efficiently selects top-performing candidates and keeps training them, avoiding training poor configurations for a long time.

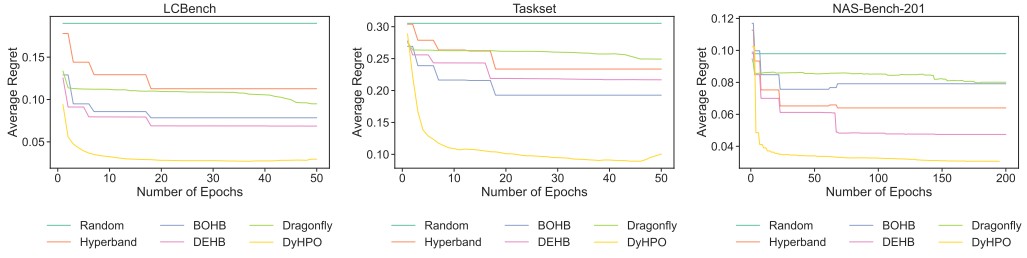

Figure 15: DYHPO spends most its budget on top-performing candidates.

## C  ON THE EFFECTIVENESS OF DYHPO

DYHPO effectively explores the search space and identifies promising candidates. This is visualized in Figure 14 in which we plot the precision of each method for different considered budgets. The precision at an epoch $i$ is defined as the number of top 1% candidates trained for at least $i$ epochs divided by the number of all candidates trained for at least $i$ epochs. The higher the precision, the less irrelevant candidates were considered and the less computational resources were wasted. For small budgets, the precision is low since DYHPO spends budget to consider some candidates but then promising candidates are successfully identified and the precision quickly increases. For LCBench and Taskset, all other methods dedicate significantly more resources to irrelevant candidates which explains why DYHPO finds good candidates faster. For NAS-Bench-201, DEHB can match the precision but only at a later stage. Simply put, the baselines select much more "poor" configurations (i.e. outside the top 1% performers) compared to our method DYHPO.

This argument is further supported by Figure 15 where we visualize the **average** regret of **all** the candidates trained for at least the specified number of epochs in the x-axis. In contrast to the regret plots in Section 6, here we do not show the regret of the **best** configuration, but the mean regret of **all** the selected configurations. The analysis deduces a similar finding as in Figure 14 above. Our method DYHPO selects highly more qualitative hyperparameter configurations than all the baselines.

## D  PROMOTION OF POOR PERFORMING CANDIDATES

An interesting property of multi-fidelity HPO is the phenomenon of poor rank correlations among the validation performance of candidates at different budgets. In other words, a configuration that achieves a poor performance at a small budget, might perform strongly at a larger budget. For instance, a well-regularized neural network will converge slower than an un-regularized network in the early optimization epochs, but eventually performs better when converged. We analyze this phenomenon and report the respective results in Figure 16. In this experiment we measure the percentage of "good" configurations at a particular budget, that were "bad" performers in at least one of the smaller budgets. We define a "good" performance at a budget B, when a configuration achieves a validation accuracy ranked among the top 1/3 compared to the validation accuracies of all the other configurations that were run until that budget B. Similarly a "bad" performance at

a budget B represents a configuration whose validation accuracy belongs to the bottom 2/3 of all configurations run at that budget B.

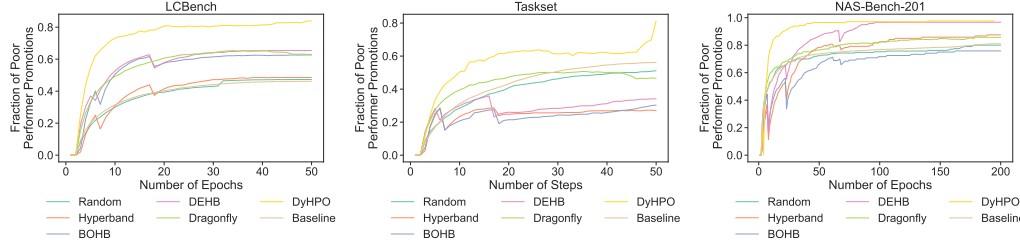

Figure 16: Percentage of configuration i) belonging to the top 1/3 configurations at a given budget, and ii) that were in the bottom 2/3 of configurations at one of the previous budgets. Here the budget is represented by the number of steps or epochs.

In Figure 16 we analyze the percentage of "good" configurations at each budget denoted by the x-axis, that were "bad" performers in at least one of the lower budgets. Such a metric is a proxy for the degree of the promotion of "bad" configurations towards higher budgets. We present the analysis for all the competing methods of our experimental protocol from Section 6. We have additionally included the ground-truth line annotated as "Baseline", which represents the fraction of past poor performers among all the feasible configurations in the search space. In contrast, the respective methods compute the fraction of promotions only among the configurations that those methods have considered (i.e. selected within their HPO trials) until the budget indicated by the x-axis. We see that in all the search spaces LCBench, TaskSet and NASBench-201 there is a high degree of "good" configurations that were "bad" at a previous budget, with fractions of the ground-truth "Baseline" varying from ca. 40% in LCBench, up to ca. 70% in the NASBench-201 datasets.

On the other hand, the analysis demonstrates that our method DYHPO has promoted more "good" configurations that were "bad" in a lower budget, compared to all the rival methods. In particular, ca. 80% of selected configurations at the datasets from the LCBench benchmark were "bad" performers at a lower budget, while in the case of NASBench-201 this fraction approaches the level of 95%.

