# OpenReview forum: "Dynamic and Efficient Gray-Box Hyperparameter Optimization for Deep Learning"
_ICLR.cc/2022/Conference — ICLR 2022 Submitted_

### Official Review · Reviewer_RUAE · 2021-10-26

**Correctness:** 2
**Technical Novelty And Significance:** 2
**Empirical Novelty And Significance:** 2
**Recommendation:** 3
**Confidence:** 5

**Main Review:**

A potential strength of this paper is the proposal of a novel surrogate model for learning-curve data which, despite involving a neural network, seems to be operational on just the data observed during a single HPO experiment. There is a lot of prior work proposing learning-curve surrogates (see below), some cited here, but most are either quite simple (multi-task GP) or require "warmstarting" on data from previous HPO experiments. Having said that, I could not find any mentioning of this point, and I am really curious about the authors explaining how their "deep GP" model can be trained just on the very limited data observed during a single HPO experiment. For an expensive tuning problem, you probably have 20-40 configurations, most of which do not run for many epochs. And even if some configurations run for many epochs, learning curve data is exceedingly noisy.

The details really matter here. With a standard BO surrogate, I just need to refit the GP hyperparameters now and then, which can easily be done even for little data. With DyHPO, you need to presumably update a deep kernel, i.e. re-train a neural network. The paper does not say how this is done in a fully automated fashion. Is training started from scratch, or from the last recent weights? The first is expensive, while the latter is prone to get stuck at the previous solution and ignores the new data. How long do you re-train? Do you re-train after getting each new observation? While using complex (deep neural) surrogate models in BO is an obvious idea, many previous trials have failed, because complex NN models are just not easy/fast to update as part of sequential decision making, and in any case cannot be fit robustly to very small datasets. This fact has been clearly spelled out, for example in [7] for the closely related problem of bandit optimization. I'd be personally really surprised if this work was different and solved these difficult issues, but would be willing to give benefit of doubt, if a lot more information was provided here how the authors pulled it off. As it stands, the authors do not even mention there could be issues here.

The most obvious weakness of this paper is that very relevant prior work is ignored, namely on asynchronous multi-fidelity. Most prominently, ASHA [1] is well known and implemented in Ray Tune [3] or AutoGluon [4]. The baselines compared to against in this paper (Hyperband, BOHB) are all synchronous (and quite dated by now), meaning that many trials need to run to a certain level until another decision is taken. If you force methods to be synchronous, this puts them at a disadvantage. They need to delay decisions until some rung is completely filled, which delays decisions and slows them down. This is explained in the ASHA paper [1]. It is well known that for large scale multi-fidelity HPO, asynchronous scheduling works much better than synchronous, see for example the comparisons in [2]. Algorithms like ASHA are behind commercial automated tuning services [5]. It is quite astonishing that part of the research community is still considering synchronous methods like Hyperband or BOHB the state of the art.

For example, the paper claims that it is a new idea that DyHPO "never discards a configuration". That is precisely what ASHA [1] does as well (known as pause-and-resume scheduling), and what Freeze-Thaw BO suggested long ago. I'd be surprised if a well configured ASHA method (available in Ray Tune [3]) would not be competitive or beat the approach suggested here, despite not requiring a complex surrogate. When doing such comparisons, it is important to also take decision time into account, because updating a surrogate model can be very expensive. In DyHPO, this likely means re-running MLP fitting, which is probably really expensive.

I also find the motivation as to why DyHPO works better than previous methods unconvincing. The authors claim that rank correlations between early evaluations (at few epochs) and late ones are poor. In my experience, this is just not the case, these correlations are in the majority pretty good, which is exactly why multi-fidelity methods work very well for DNN tuning. Sure, there are examples such as regularization, but the question is whether that matters. The authors should provide numerical evidence for such a claim. Now, even if these correlations are poor, it is not clear to me why DyHPO could do anything about that. Learning curve prediction is just hard, because by far most of the data is from early evaluations, but you are interested in late performance, so you need to extrapolate. Why would some vanilla deep kernel be good at that? The only way to really know about certain anti-correlations that can be exploited, is to either fit models to data from past HPO experiments (which DyHPO does not do), or to built the knowledge into the model (which they don't do either). Just because an NN is involved, does not mean it will do magic for you. The reason why DyHPO works better than competitors here is that it is asynchronous, but the others are synchronous, so at a disadvantage. Also, the reason why model-based HPO is better than random search based methods (like Hyperband) is mostly because the latter cannot exploit: they need to draw new configs always at random.

The experiments are pretty underwhelming. Apart from most relevant baselines missing (DyHPO is asynchronous, all competitors are synchronous), the curves are also not very meaningful, because the x axis is number of epochs instead of wall-clock time. DyHPO needs to update a complex surrogate model, including retraining a neural network, and the costs for doing that have to be taken into account. All experiments are also sequential, no parallel evaluations are used (this could easily be done by using Ray Tune [3]), again this falls far short of the current state of the art in automatic tuning of large neural models (e.g., methods like ASHA or PBT).

Finally, there are quite some works on using complex surrogates to model learning curves in the context of HPO, for example [5]. It is not clear why this was not compared against, as code is available. The work of Wistuba and Grabocka is cited, which proposed deep kernel surrogates before (so this paper, against their claim, is not the first to do this in the context of multi-fidelity), and in fact Perrone etal (2018), cited here, did this even earlier, just not in the context of learning curve data. The paper of Wistuba is quite careful in explaining why a complex surrogate cannot be trained robustly on the data from a single experiment, and proposes an algorithm to warmstart from past data. It is dismissed here (as competitor) for doing so, but as I said above, I am not sure how DyHPO solves the apparent issue that complex NNs cannot be trained on the small amount of data observed in HPO.

[1] ASHA: https://arxiv.org/abs/1810.05934
[2] MOBSTER: https://arxiv.org/abs/2003.10865
[3] Ray Tune: https://docs.ray.io/en/latest/tune/index.html
[4] AutoGluon: https://auto.gluon.ai/stable/index.html
[5] https://www.determined.ai/
[6] https://openreview.net/forum?id=S11KBYclx
[7] https://arxiv.org/abs/1802.09127


**Summary Of The Paper:**

This paper is concerned with multi-fidelity HPO (the authors call this "grey-box", which is a non-standard term). They propose a surrogate model for learning curve data (e.g., metric values at each epoch) based on a deep GP kernel. Different to most previous work on synchronous multi-fidelity HPO, they decide for each running trial when it should be continued. Experiments are presented, where the method is compared to a range of synchronous HPO baselines. The experiment are fairly small-scale and do not use parallel evaluations.


**Summary Of The Review:**

The paper may have some merits in suggesting a "deep kernel" surrogate model which, although quite related to previous work, is stated to work even if just fitted on the small amount of data from a single experiment, in an online fashion. However, this has been tried several times before with little success, and details explaining why the current approach should work are missing.

The proposed method uses asynchronous scheduling (much like Freeze-Thaw), but is compared against synchronous scheduling baselines, which have a major disadvantage. Comparisons to SotA methods like ASHA (or PBT) are missing, these are not cited. There is also quite a range of prior work on learning curve modeling for HPO, which is not compared against. Open source code for doing a better comparison is publicly available (for example, Ray Tune).

Experiments are smallish scale, mostly on tabulated benchmarks, and again are not close to what is possible today with parallel computation. Compared to missing alternatives like ASHA, the proposed method is fairly complex and quite likely rather non-robust to handle. For example, it requires retraining a neural network model each time a bit of new data is obtained, which is very difficult to do.

---

> ### Author Response · Authors · 2021-11-10
> **Reply to Reviewer RUAE - Part 1**
>
> Thanks a lot for your extensive review and the points you are raising.
>
> **On Limited Noisy Observations:**
>
> We highlight the fact that training Deep GP surrogates for limited observations has been already shown to be feasible by FSBO [1] and is the current state-of-the-art in transfer learning according to the recent experimental study released with the HPO-B benchmark [2]. Additionally, the benchmark experiment shows that a deep GP (standard HPO, not a transfer setup) outperforms a standard GP. These results contradict the perception that training a regularized deep GP on a “limited” number of observations is not feasible.
>
> The reviewer additionally raises the concern that the learning curve data is noisy. However, that behavior is handled directly by our acquisition function, i.e. Expected Improvement of Section 5.2, which uses both the mean and the uncertainty conditional to a budget/fidelity. In that sense, the noise of a learning curve will yield a higher posterior variance by the GP, which is handled directly in the exploration mechanism of the Expected Improvement acquisition.
>
> **On the procedure of fitting a Deep GP:**
>
> Regarding the question of how to efficiently fit a Deep GP surrogate, the reviewer is right that training the GP kernel parameters adds a very small overhead to the overall HPO runtime. However, fitting a deep GP with a two-layer kernel on observations of a few hyperparameter configurations is way faster than actually evaluating the performance of a configuration, i.e. fitting the parameters of the actual deep learning model. We consider the overhead of the surrogate fitting to be a negligible constant in line with prior experimental protocols of the prior work. For example on the largest search space with the largest number of hyperparameters (NASBench201), it takes only 4 secs to refit our deep GP from a history of 100 configurations evaluated at diverse budgets (i.e. only up to 4 secs of overhead for each configuration we sequentially apply).
>
> We further highlight that the reviewer’s criticism on the lack of details on the optimization procedure of fitting the deep GP was already answered in Appendix A.4. Please let us know if there are still questions remaining after reading the description in the Appendix.
>
> **On rank correlation and how DyHPO handles it**
>
> The suboptimal correlation of successive halving techniques arises from the fact that configurations are discarded at low fidelities, even if they end up performing stronger with larger fidelities.
> Regarding the question of how we handle rank correlations, the answer is that we never discard a configuration after seeing a poor performance at low fidelity. Consider a configuration for which you have seen a lot of epochs already. In that case, the deep GP will estimate a high validation accuracy for the next epoch (as the method has already converged), BUT the variance will be low as this configuration has converged and is unlikely to improve significantly after one epoch/step. In contrast, for a configuration where we have seen only a few epochs of its learning curve, our deep GP will have a low posterior mean, BUT a high posterior variance, as the configuration has not converged (i.e. not enough observations for the GP to reduce the variance). Therefore, at some point in time, the Expected Improvement acquisition (which combines the posterior mean and uncertainty) will stop training converged configurations with low uncertainty, and continue training non-converged configurations with higher uncertainty. The reason why our method achieves state-of-the-art lies in the elegant dynamic switching between posterior means (high validation accuracy) and posterior variance (non-converged configurations), which is the intrinsic mechanism of Bayesian Optimization and the Expected Improvement acquisition.

---

> > ### Author Response · Authors · 2021-11-10
> > **Reply to Reviewer RUAE - Part 2**
> >
> > **On synchronous vs asynchronous:**
> >
> > The reviewer highlights the argument that our method DyHPO outperforms the baselines, only because it is asynchronous, while the remaining baselines are synchronous. We correct the reviewer, because our method is synchronous, as we have a standard Bayesian Optimization setup that decides the configuration to train next in a sequential manner. No configuration is run in parallel in our method.
> >
> > As a result, given that our method is sequential and not asynchronous/parallel, it is a fair comparison against the other sequential baselines. In addition, all the experiments abide by a sequential HPO modality where the total sequential HPO runtime budget is measured.
> >
> > We agree with the reviewer that asynchronous HPO techniques are important for utilizing large cluster resources, however, in this paper, we do not propose an asynchronous/parallel HPO method. Parallel HPO is a different problem definition compared to the sequential HPO that we are addressing. Both paradigms have their own merits. A sequential HPO approach is relevant in the case of training deep models on a single server with multiple GPUs (a common use case for many small companies and research labs that do not own large clusters for deploying parallel HPO methods). For an analogy of this line of reasoning, all the papers proposing novel optimization algorithms with experiments on 1 server would have to be rejected if they do not converge as quickly as a distributed SDG that runs in a cluster of 100 servers, which we believe to be a principally flawed argument.
> >
> > In this context, a comparison to parallel HPO baselines like ASHA or PBT is out of scope for the sequential HPO problem definition, but a nice recommendation for us to consider expanding our work towards parallelization in the future, once this paper is accepted.
> >
> > **On novelty, modeling learning curves:**
> >
> > The author highlights that there are many papers that model learning curves, citing [3]. However, we were not able to identify which paper the reviewer is pointing at, as [3] is a commercial platform instead of a publication.
> >
> > We stress that modeling learning curves is only one component of building an HPO surrogate, however, for a multi-fidelity HPO method you also need the concept of deciding what configuration to run, and for what budget.
> >
> > We would be glad if the reviewer can share any method that models learning curves in the context of sequential multi-fidelity HPO techniques?
> >
> > **On baselines:**
> >
> > The reviewer believes our method has an unfair advantage against baselines because it is an asynchronous method, while the baselines are synchronous. As a result, the reviewer suggests we compare to other asynchronous baselines.
> > We clarify again that our method is a sequential HPO approach and not an asynchronous one. As a result, the criticism is not well-grounded, therefore, we invite the reviewer to reconsider her/his criticism.
> > Furthermore, the reviewer points out a related work [1] as a prior work that builds complex surrogates in the context of multi-fidelity. We are familiar with FSBO [1] and it is not a multi-fidelity approach, instead, it is a black-box HPO method in a transfer-learning setup. Furthermore, the more recent paper [2] of the very same authors from [1] demonstrates that deep GPs can also be efficiently trained from limited observations in the non-transfer setup (method “Deep Kernel GP” in Figures 1,4,5 at [2]), i.e. without meta-initialization of the deep GP parameters. The argument that deep GPs cannot be trained from limited observations and therefore are not suitable for HPO is a perception that is contradicted by the empirical findings of prior work [2].
> > In addition, we are familiar with the other mentioned related work from (Perrone et al., 2018), that meta-learns Bayesian linear regression surrogates for a black-box HPO setup (That is quite a different method to ours in multiple directions: i) transfer learning HPO vs. standard non-transfer HPO, ii) black-box vs. multi-fidelity HPO, iii) Bayesian linear regression vs. deep GP surrogates, iv) no budget/learning curve modeling). We are curious to hear what is the relation the reviewer implies between our work and (Perrone et al., 2018)?

---

> > > ### Author Response · Authors · 2021-11-10
> > > **Reply to Reviewer RUAE - Part 3**
> > >
> > > **Summary:**
> > >
> > > We clarified that the major criticism of the reviewer in terms of the unfair disadvantage of our method was a misunderstanding. Our paper is sequential and not asynchronous as the reviewer characterized it, therefore, the suggested missing asynchronous baselines are out of the scope of our paper.
> > >
> > > Furthermore, the reviewer also raised a criticism that deep GPs cannot be efficiently trained from a small number of evaluations, which is contradicted by the findings of prior work [2], where deep GPs are shown to achieve the new state-of-the-art in both non-transfer and transfer black-box HPO on a newly-released HPO benchmark.
> > >
> > > Also the reviewer hypothesizes that fitting a deep GP is too slow and adds a significant overhead to the overall HPO runtime. This is not correct, as fitting our deep GPs in the considered number of evaluations is very fast. For example, on the largest search space with the largest number of hyperparameters (NASBench201), it only takes only ca. 4 secs to train our deep GP from a history of 100 evaluated configurations at diverse learning curve lengths.
> > >
> > > In addition, the reviewer criticizes our novelty by stating that there are “quite some works on using complex surrogates to model learning curves”. However, out of 3 pointed related works, one of them is a commercial platform whose underlying method is not clearly distinguishable [3]. In addition, the two other prior papers the reviewer raises are not multi-fidelity HPO methods and solve a different black-box HPO problem definition (see On Baselines above).
> > >
> > > As a result, we would be glad to hear if the reviewer still has a rejection reason, that would like to be clarified in order to increase the score to an acceptance.
> > >
> > > [1] FSBO https://openreview.net/forum?id=bJxgv5C3sYc
> > > [2] HPO-B https://openreview.net/forum?id=O24OhmqpJtP
> > > [3] https://www.determined.ai/

---

> > ### Comment · Reviewer_RUAE · 2021-11-19
> > **Reply to Author Feedback (all parts)**
> >
> > Here is my reply to the lengthy 3-part response of the authors.
> >
> > First, I decided that my initial assessment was a bit harsh, and some arguments of the authors convinced me (below), so I will increase my score and slightly lower my confidence. However, I still have quite some "rejection reasons" that simply have not been addressed, and so I still think this paper should not get accepted.
> >
> > * Limited noisy observations, fitting Deep GP
> >
> > I agree with the authors that fitting the deep GP after each update is probably quite fast, because there is so little data to fit. I read Appendix A.4 now. The big question to me is why fitting a NN model with 1000s of parameters on the learning curve data from maybe 20-30 configs is a robust procedure? Papers like Wistuba et al (cited, as FSBO [1]) point out that this is very difficult at the least, unless one has data from previous experiments.
> >
> > Here, the number of configs matters much more than the total number of points, the latter I could always increase by sampling the learning curve more often (e.g., several times per epoch), since it is configs that the algorithm makes decisions on.
> >
> > Appendix A.4 says that "For every new data point, we start training the GP with its old parameters to reduce the required effort for training." Such incremental training approaches are typically quite unsucessful, in that the NN first totally overfits the data at the beginning (2-3 configs), and if you do not restart from random weights later on, you never really get out of this rapidly. Also, the authors claim to optimize the marginal likelihood with an optimizer using mini-batches. The ML does not decompose as a sum over batches, so I suppose some kind of variational bound is used here. With this little data, it is odd that a batch gradient optimizer like L-BFGS was not chosen here.
> >
> > All in all, I maintain the authors do not really address the point while their method of incremental training of a NN on tiny amounts of data works robustly, when many previous approaches did not manage to do this ([7] for bandits, quite some more data; [8] for BNNs; also FSBO).
> >
> > It is quite remarkable that in their response, they cite FSBO [1] as an example why fitting a DeepGP to very little data can work as part of HPO, while the very same paper is not doing that, but instead relies on data from past experiments, and for exactly this reason, the authors here do not compare against FSBO. The way I read the FSBO paper was that they use past experiment data exactly because things do not work if you do not.
> >
> > * Rank correlations
> >
> > Like another review points out, the claim that rank correlations are poor in general for such DNN tuning problems is overblown, and the authors do nothing to justify it. Also, ASHA and Freeze Thaw are using pause-resume scheduling, so do not stop configs once and for all, but may restart them later. While the motivation why the present DeepGP method could do a better job at pause and resume decisions sounds nice, it is simply a hunch of the authors, without any experimental justification. In particular since the DeepGP model is trained on such a tiny amount of data.
> >
> > * Sync versus async, baselines
> >
> > Sync and async have not much to do with parallel computing. In a sync method, many evaluations have to get to a certain level before a (group) decision is made. In Hyperband, all configs in a rung need to be run to the next level before a stop/continue decision is then made for all of them. In an async method, decisions about each config can be made without having to wait for others. The method proposed here is exactly like this, so it is async. ASHA can perfectly be run with a single worker, if you like to do that, so can be Freeze Thaw BO.
> >
> > I maintain that a comparison with ASHA or any other async method is missing here, the authors can run it with a single worker if they like. It is my strong guess that this would explain the differences to older sync work like Hyperband and BOHB shown here.
> >
> > * Novelty, modelling learning curves
> >
> > There is a lot of prior work on modelling learning curves and then (of course) using the model to make stop/go decisions, for example [8, 9], I could find more. Also, Freeze Thaw BO is an example, even though they use a simple kernel function. [8] uses an elaborate Bayesian NN.
> >
> >
> > [8] Klein et al: Learning Curve Prediction with Bayesian Neural Networks (https://openreview.net/forum?id=S11KBYclx)
> >
> > [9] Domhan et al: Speeding up automatic hyperparameter optimization of deep neural networks by extrapolation of learning curves, IJCAI 2015

---

> > > ### Author Response · Authors · 2021-11-20
> > > **Second rebuttal to Reviewer RUAE (Part 1)**
> > >
> > > Thanks a lot for finding time to read through our initial rebuttal. We appreciate your comments and reply to them as follows:
> > >
> > > **On the limited noise observations, fitting Deep GP**
> > >
> > > We would like to re-emphasize that the related works [1,2] do train deep GPs on limited observations, in two modalities:
> > >
> > > - Transfer learning for black-box HPO: FSBO [1] initializes the deep GP network parameters with the converged weights of a meta-learned Deep GP (transfer setup). However, for a new task/dataset the GP is trained further on the few configurations of the new task, exactly as we do.
> > > - Normal black-box HPO: Deep Kernel GP [2] is a variant of FSBO [1] that is initialized randomly (as in our case, not from the converged weights of a meta-learning procedure). Then the Deep Kernel GP from [2] is trained on the few configurations of a given task/dataset, exactly as we do.
> > >
> > > In contrast to FSBO we do not tackle transfer-learning, i.e. initialize the surrogate from evaluations on past datasets. Furthermore, we target multi-fidelity HPO, not black-box HPO as in [1,2]. However, the procedure of training deep GPs on limited configurations in [1,2] and our paper is identical. The only points we do differently are:
> > >
> > > - in our case the surrogate is adopted for the multi-fidelity case by fusing budget and learning curve information;
> > > - the acquisition function and the BO procedure is adopted to the multi-fidelity HPO setup.
> > >
> > > We believe that the black-box HPO empirical evidence from [1,2] on 196 datasets, as well as our multi-fidelity HPO results on 50 datasets, clearly demonstrate that fitting deep GPs on limited observations is feasible, works in practice, and leads to state-of-the-art HPO surrogates.
> > >
> > > The relation between the parameters of the prediction model and the number of labeled instances can only be answered empirically. E.g. CIFAR10 has 50K labeled training instances, and yet, the state-of-the-art architectures have millions of parameters [5]. Our position is: Only the empirical evidence demonstrates whether a method works, not the apriori judgment based on the number of parameters.
> > >
> > > To sum up, we have shown strong empirical evidence on the fact that fitting deep GPs for HPO works great empirically on 50 datasets from three diverse search spaces, a finding which is in line with large-scale experiments from prior work (176 search spaces, 196 datasets) [1,2].
> > >
> > > If the reviewer is still suspicious of the matter, we welcome empirical counter-evidence to support the criticism.
> > >
> > > **Rank correlations**
> > >
> > > We are presenting an answer to this concern as a separate comment (see above) because multiple reviewers raised the point. Please see the post with the title "On poor correlation and the effectiveness of our method" here https://openreview.net/forum?id=aBAgwom5pTn&noteId=lXv7bQwUdRi.
> > >
> > > **Sync vs. async baselines**
> > >
> > > The reviewer clarifies the definition of async methods as HPO methods that do not wait for configurations to complete a fixed initial budget (e.g. contrary to the concept of the initial rung budget in Hyperband). Furthermore, the reviewer criticizes again our experimental protocol, stating that "... any other async method is missing here".
> > >
> > > We kindly correct the reviewer's comment that we do not compare to any async baseline. Following your definition of async methods, DragonFly is an async state-of-the-art method for multi-fidelity HPO [6,7]. We have compared against DragonFly and outperformed it significantly. Does this comparison already address your concern?
> > >
> > > **Novelty, modeling learning curves**
> > >
> > > We would like to reiterate the response to reviewer ty1k that raised the same concern. Our clarification was: "Learning curve prediction is concerned with forecasting the learning curve but not with hyperparameter optimization. A multi-fidelity HPO method can incorporate a learning curve model, but also needs a mechanism of deciding which configuration to try next and for what budget. Therefore, learning curve estimation papers are not by default multi-fidelity HPO methods." The suggested papers [8,9] are not multi-fidelity HPO methods, but simply learning curve estimators, because they do not answer the question "which configuration to select next and for what budget to run it?".
> > >
> > > Freeze-Thaw is a missing multi-fidelity baseline, you are right on that one! However, we could not find a reliable public implementation for this method. We did run the unofficial implementation from https://github.com/mrenoon/datafreezethaw, but the results were quite poor, and we were not sure whether the implementation is correct. On the other hand, we have already compared to the most accurate multi-fidelity HPO method that we are aware of. As a supporting argument, prior work indicates that Freeze-Thaw is outperformed by Hyperband [10], while we outperform Hyperband significantly.
> > >
> > > Please let us know whether your rejection reasons are clarified.
> > >
> > > (References continue in the comment below)

---

> > > > ### Author Response · Authors · 2021-11-20
> > > > **Second rebuttal to Reviewer RUAE (Part 2)**
> > > >
> > > > References:
> > > >
> > > > [1] FSBO https://openreview.net/forum?id=bJxgv5C3sYc
> > > >
> > > > [2] HPO-B https://openreview.net/forum?id=O24OhmqpJtP
> > > >
> > > > [3] Klein et al: Learning Curve Prediction with Bayesian Neural Networks (https://openreview.net/forum?id=S11KBYclx)
> > > >
> > > > [4] Domhan et al: Speeding up automatic hyperparameter optimization of deep neural networks by extrapolation of learning curves, IJCAI 2015
> > > >
> > > > [5] Dong et al., Searching for A Robust Neural Architecture in Four GPU Hours, CVPR 2019, https://arxiv.org/pdf/1910.04465.pdf
> > > >
> > > > [6] Kandasamy et al.,  Multi-fidelity Bayesian Optimisation with Continuous Approximations, ICML2017 https://proceedings.mlr.press/v70/kandasamy17a/kandasamy17a.pdf
> > > >
> > > > [7] Kandasamy et al., Tuning Hyperparameters without Grad Students: Scalable and Robust Bayesian Optimisation with Dragonfly, JMLR 2020 https://people.eecs.berkeley.edu/~kandasamy/pubs/kandasamyJMLR20dragonfly.pdf
> > > >
> > > > [8] Klein et al: Learning Curve Prediction with Bayesian Neural Networks (https://openreview.net/forum?id=S11KBYclx)
> > > >
> > > > [9] Domhan et al: Speeding up automatic hyperparameter optimization of deep neural networks by extrapolation of learning curves, IJCAI 2015
> > > >
> > > > [10] Lu et al., Hyper-parameter Tuning under a Budget Constraint ICJAI 2019, https://www.ijcai.org/proceedings/2019/0796.pdf

---

> ### Author Response · Authors · 2021-11-30
> **Added new baseline ASHA**
>
> Please let us kindly inform you that we added ASHA [1] to our experiments, the baseline you indicated was missing.
>
> The extended results are accessible in our previous post here https://openreview.net/forum?id=aBAgwom5pTn&noteId=MlhgfpdLL9b
>
> [1] ASHA: https://arxiv.org/abs/1810.05934

---

### Official Review · Reviewer_cMi3 · 2021-11-03

**Correctness:** 1
**Technical Novelty And Significance:** 1
**Empirical Novelty And Significance:** 1
**Recommendation:** 3
**Confidence:** 4

**Main Review:**

major concerns

1. There are several parts of this paper that are not clearly related to similar studies.
    e.g.
	- multi-fidelity BO with deep models
		- Li+ "Multi-Fidelity Bayesian Optimization via Deep Neural Networks" NeurISP2020
	- EI for multi-fidelity BO
		- Picheny+ "Quantile-Based Optimization of Noisy Computer Experiments with Tunable Precision" Technometrics, 55(1):2-13
		- Lam+ "Multifidelity Optimization using Statistical Surrogate Modeling for Non-Hierarchical Information Sources" 56th AIAA/ASCE/AHS/ASC Structures, Structural Dynamics, and Materials Conference. 2015.

2. There is a lack of explanation about the architecture of the deep kernel (why this architecture, what makes this architecture effective for multi-fidelity optimization, etc.). It looks to be just a presentation of a kernel architecture that happens to work well.

3. It is not appropriate to plot only the average riglet in the experimental results, so the variance should be plotted as well (if it is difficult to plot, it can be reported separately).

4. Although the effectiveness of the multitask kernel is evaluated in ablation study, modeling using multi-task kernels in multi-fidelity optimization has already become popular and has been evaluated in various studies (e.g., https://arxiv.org/abs/1406.3896, https://arxiv.org/abs/1605.07079, https://arxiv.org/abs/1903.04703). Rather, what we should consider is what parts of the deep kernel structure are effective and why.

**Summary Of The Paper:**

This paper proposes a gray-box optimization method for hyperparameter optimization of deep neural network models. In order to deal with the different budgets available for training NNs in the framework of multi-fidelity optimization, the proposed method uses a multi-task Gaussian process modeling that simultaneously measures the similarity between not only the inputs x but also the outputs y trained with different budgets. In particular, the multi-task Gaussian process model is constructed using a deep kernel with a feature extractor instead of the existing kernel function. The performance of the proposed method is evaluated by experiments on three types of neural nets: MLP, RNN, and CNN. MLP and RNN are treated as usual hyperparameter optimization in 7 and 8 dimensions, respectively, while CNN is treated as a rewrite of NAS as hyperparameter optimization.

**Summary Of The Review:**

I cannot support the acceptance of this paper due to insufficient evaluation of the novelty and the effectiveness of the proposed method.

---

> ### Author Response · Authors · 2021-11-10
> **Reply to Reviewer cMi3 - Part 1**
>
> Thanks a lot for the points you are raising.
>
> **On novelty:**
>
> Please notice that in Section 3 we did describe the related work on similar approaches for Multi-fidelity Bayesian Optimization, where we also cite the paper you are referring to. At the end of Section 3, we also highlighted how our method delineates from these other works as: “In contrast to prior work, we propose the first deep kernel GP for multi-fidelity HPO that is able to capture the learning dynamics across fidelities/budgets, combined with an acquisition function that is tailored for the gray-box setup.” None of the prior work captures the dynamics of the learning curves across fidelities as we do through our deep GP, that fuses the hyperparameter configuration, the fidelity/budget information, and the learning curve via a parametric end-to-end surrogate.
>
> In that context, we actually compared against the most popular related work in terms of multi-fidelity BO [1,2], which is also supported by a recent library DragonFly [JMLR 2020]. Regarding the acquisition function, please notice that the baseline [2] already has a dedicated multi-fidelity acquisition function. In all the experiments, our method significantly outperforms the baseline [1,2] with its multi-fidelity BO surrogate and acquisition.
>
> We will work towards adding the second baseline you mentioned [3] and report results with it as soon as ready.
>
> **On the architecture of the proposed surrogate:**
>
> We are sorry if the architecture feels like an ad-hoc architecture without proper motivation. The deep GP surrogates are the new state-of-the-art surrogate for transfer and non-transfer black-box HPO and have recently gathered momentum in the community [4,5]. The deep GP we are proposing is aligned with the architecture definition of [4,5] which we have described in the related work, however, it is adopted for the multi-fidelity HPO setup.
>
> In this regard, for a multi-fidelity HPO setup, each hyperparameter configuration has three components:
> - The hyper-parameter configuration vector, i.e. values of hyper-parameters x
> - The budget information, i.e. the index j at our formalism
> - The learning curve so far, i.e. the sequence Y in our formalism
>
> We followed the most intuitional way to combine these three sources of features by fusing them through concatenation and then capturing the interaction via a neural network. As the learning curve is a sequence, not a vector, it was projected to a vector via a CNN model (we tried RNN, but CNN worked better). Then the three representation vectors (config, budget, learning curve) are concatenated and fed into a MLP network. We believe that our architecture is a standard and intuitional way of feature fusion, and we are not aware of a more direct way of combining these three sources of features for a deep GP. We would be glad to hear whether the reviewer has a suggestion on constructing the architecture differently?
>
> We will revise the paper to highlight that the architecture of a deep GP is motivated by [4,5] and adapted to capture the multi-fidelity mechanism.
>
> Nevertheless, please note that in contrast to the prior work [4,5], our deep GPs are trained for a multi-fidelity setup and do not estimate the final performance (black-box HPO), but the next budget performance (multi-fidelity HPO). In addition, we developed a new acquisition to capture the expected improvement at that particular budget for which our surrogate is making the prediction.
>
> **On baselines:**
>
> Overall, although our approach is also a “Multi-fidelity BO” method, it is quite unique in the way the surrogate is defined, in the way it is trained as a next-budget performance forecasting, and in the way the acquisition operates in the BO setup. The comparison against the state-of-the-art methods in multi-fidelity HPO, including the baseline on Multi-fidelity BO [1,2] empirically demonstrate the strong performance of our method in a large-scale protocol.
>
> Furthermore, you raised a criticism on the similarities to some of the prior work (e.g., https://arxiv.org/abs/1406.3896, https://arxiv.org/abs/1605.07079, https://arxiv.org/abs/1903.04703).
>
> The main difference from these papers is that we learn to capture the similarity of two configurations that are evaluated at different budgets (Eq. 6 in our paper). In order to capture this interaction, we need a neural network inside the GP kernel to actually learn the interactions among the configuration values, the budget, and the projection vector of the learning curves. In contrast, the other methods do not capture this interaction among configurations as their kernels are non-parametric (https://arxiv.org/pdf/1406.3896.pdf), or they rely on heuristics for the acquisition function (https://arxiv.org/pdf/1903.04703.pdf), instead of making the surrogate’s estimations contextualized on the budget information.

---

> > ### Author Response · Authors · 2021-11-10
> > **Reply to Reviewer cMi3 - Part 2**
> >
> > **On the evaluation metric:**
> >
> > The reviewer raises a suggestion on showing the variances of the regret curves.
> >
> > We point out that the methodologically correct mechanism of comparing methods across datasets are statistical-significance tests, such as Wilcoxon Signed-Rank [6], which is the recommended mechanism of measuring the statistical significance of HPO methods by recent state-of-the-art papers [5].
> >
> > In that sense, we already measured the statistical significance of our method in accordance with the best practices. Overall, we have demonstrated that we outperform recent papers published at top venues, such as HyperBand (JMLR 2018), BOHB (ICML 2018), DragonFly (JMLR 2020), DEHB (IJCAI 2021), with a large and statistically-significant margin on a large-scale experiment comprising of 50 datasets in 3 different search spaces.
> >
> > **Summary**
> >
> > Having clarified the confusion on the novelty aspects of our paper and the choice of the architecture, we would be glad to hear if there is a rejection reason you might still have, or points that you would like to be clarified in order to increase the score to acceptance?
> >
> >
> > [1] https://proceedings.mlr.press/v70/kandasamy17a/kandasamy17a.pdf
> > [2] https://people.eecs.berkeley.edu/~kandasamy/pubs/kandasamyJMLR20dragonfly.pdf
> > [3] "Multi-Fidelity Bayesian Optimization via Deep Neural Networks" NeurISP2020
> > [4] FSBO https://openreview.net/forum?id=bJxgv5C3sYc
> > [5] HPO-B https://openreview.net/forum?id=O24OhmqpJtP
> > [6] https://www.jmlr.org/papers/volume7/demsar06a/demsar06a.pdf

---

> ### Author Response · Authors · 2021-11-30
> **Added new baseline MF-DNN**
>
> Please let us kindly inform you that we added MF-DNN [1] to our experiments, the baseline you indicated was missing.
>
> The extended results are accessible in our previous post here https://openreview.net/forum?id=aBAgwom5pTn&noteId=MlhgfpdLL9b
>
> [1] Li+ "Multi-Fidelity Bayesian Optimization via Deep Neural Networks" NeurISP2020

---

### Official Review · Reviewer_oS3h · 2021-11-04

**Correctness:** 3
**Technical Novelty And Significance:** 3
**Empirical Novelty And Significance:** 3
**Recommendation:** 6
**Confidence:** 3

**Details Of Ethics Concerns:**

I do not see any ethical concerns.

**Main Review:**

Strengths:
- Multi-fidelity is very important to obtain practical HPO algorithms for deep learning.
- The proposed deep kernel accounts for the correlation between learning curves to avoid naively stopping trials to early like HB does.
- The modification proposed to account for the learning curve is fairly simple
- The experiments are quite convincing, with DyHPO outperforming other HPO algorithms almost systematically. The baselines are good, with Hyperband, BOHB and DEHB being serious multi-fidelity contenders.

Weaknesses:
- It is very unclear to me how we can guarantee that the algorithm will often resample x to make it continue. If most dimensions of the search space are real, it seems to me x will most likely be different than previous ones, leading the algorithm to never continue trials. The empirical results clearly show that this is not the case, however the explanations do not make clear why it would not be the case.
- Only figure 8 reports training time in seconds (I assume including HPO time to suggest new trials as well). With the frequent query on the algorithm (every 1 epoch) I assume there must be a significant amount of overhead, starting and stopping trials very often. Also the algorithm itself must be fairly slow compared to hyperband, with the deep kernel that requires training. It would be best to report more results on the running time of DyHPO.
- There are no clear experiments with learning curves that are best later on to synthetically show that DyHPO performs well in this case. This would be extremely valuable to support strongly that the reason why DyHPO works so well is that it indeed let the best trials train even though they do not perform well at the beginning.

**Summary Of The Paper:**

This paper present a new Bayesian Optimization algorithm that integrates a Deep Kernel over both the hyperparameters x and the fidelity budget j (typically number of epochs). It also present a slightly modified version of the expected improvement acquisition function to for the fidelity budget. The paper shows on several benchmarks that the proposed algorithm, called DyHPO is highly competitive, better performing than BOHB and DEHB on most of the benchmarks, otherwise performing similarly.

**Summary Of The Review:**

The algorithm presented in this paper is an appreciable improvement to multi-fidelity variants of Bayesian Optimization, especially because it accounts for the correlation between the learning curves and avoid relying to strongly on low fidelity to make hard decisions on trials to stop or continue training. The experiments are convincing, they are broad and compare good baselines. The paper lacks important details in my opinion with respect to the optimization of the expected improvement and how it ensures a good fraction of the trials continue training. It also lacks analysis of the execution time of the algorithm and more explicit experiments showing it can avoid stopping good trials that progress slowly in the first epochs. I consider the work good enough for publication, but it could benefit from some clarifications and additional analysis.

---

> ### Author Response · Authors · 2021-11-20
> **Rebuttal to Reviewer oS3h**
>
> Thank you very much for your valuable comments and raised points.
>
> **On how configurations are resumed:**
>
> The configurations are resumed as a result of the operating principle of Bayesian Optimization and the uncertainty modeling of the Expected Improvement acquisition. At the moment the configuration leads to a converging learning curve the uncertainty drops, then the Expected Improvement drops, too. At that point, another configuration with a higher posterior GP uncertainty and a higher Expected Improvement can be selected. Once the uncertainty and Expected Improvement of the second configuration also decrease (as it converges after some epochs), then again the Expected Improvement of the first might be numerically higher again (e.g. because it has a larger posterior GP mean). Long story short, at different points in the HPO procedure the posterior variance of the same hyperparameter configuration changes/decreases. Therefore, its Expected Improvement decreases and gives room for the BO procedure to (re-)select other configurations.
>
> **On the additional overhead time of fitting the deep GP:**
>
> In order to show that the overhead of DyHPO is not significant, we present new LCBench experiments and aggregated wall-clock results in Figure 1. The detailed wall-clock results of each dataset are in Figures 12 and 13. Furthermore, the wall-clock results were already included for the datasets of the NASBench201 tasks. Regarding the TaskSet problems, the benchmark does not offer the runtime for an evaluation, therefore we use the number of steps as a runtime proxy. The message of the new experiments is that the overhead does not change the state-of-the-art outcome of our method DyHPO compared to rivaling approaches.
>
> **On the poor correlation of configurations:**
>
> We analyze how well can DyHPO resume configurations that are good at a budget, but sub-optimal at previous budgets, in the new experiment in Appendix D, Figure 16 in the updated manuscript. Because multiple reviewers raised that point, we posted a separate answer here https://openreview.net/forum?id=aBAgwom5pTn&noteId=lXv7bQwUdRi

---

### Official Review · Reviewer_ty1k · 2021-11-04

**Correctness:** 3
**Technical Novelty And Significance:** 2
**Empirical Novelty And Significance:** 3
**Recommendation:** 5
**Confidence:** 3

**Main Review:**

Pros:
- Incorporating learning curve dynamics into the surrogate model is well motivated and supported by the ablation study on the NAS-Bench 201 dataset.
- Extensive experimental results have been provided in terms of tabular datasets, NLP tasks, and NAS.

Cons:
- Predicting learning curves is not new for HPO as it has been well explored by previous works like [1]. While the proposed method tries to involve the budget information for modeling curve dynamics, the technical novelty of this work is still somewhat limited since it seems like a direct combination between [Wilson et al., 2016] and [Kandasamy et al., 2017].
- The multi-fidelity acquisition function is not well supported by the ablation study. What is the comparison result between DYHPO and DYHPO w/o MF?
- Some necessary baselines are missing in the current experiment, such as [1] and [Wilson et al., 2016].

[1] Learning Curve Prediction with Bayesian Neural Networks, ICLR’17.


**Summary Of The Paper:**

A gray-box hyperparameter optimization framework has been developed based on a multi-fidelity acquisition function and a surrogate model incorporated with learning curve dynamics. The proposed method was built on top of deep kernel learning [Wilson et al., 2016] and multi-fidelity Bayesian optimization [Kandasamy et al., 2017]. Experimental results on three different settings were provided in optimizing hyperparameters for MLP, RNN, and CNN, respectively.

**Summary Of The Review:**

Overall, the paper is easy to follow and well-motivated. While some ablated models and baselines are missing, the experimental results are comprehensive and seem to be solid. The main concern of this work is the lack of technical novelty compared with existing works.

---

> ### Author Response · Authors · 2021-11-12
> **Reply to reviewer ty1k**
>
> We thank you for the interesting questions and comments.
>
> **On learning curves and related work**
>
> Learning curve prediction is different from our work. Learning curve prediction is concerned with forecasting the learning curve but not with hyperparameter optimization. A multi-fidelity HPO method can incorporate a learning curve model, but also needs a mechanism of deciding which configuration to try next and for what budget. Therefore, learning curve estimation papers are not by default multi-fidelity HPO methods. We would be glad if the reviewer can point out missing related work that incorporates the learning curve dynamics into a multi-fidelity HPO method?
>
> In addition, we highlight that our paper is not an extension of combining [1] and [2]. We assume the reviewer means that we use [2] as a learning curve regressor (deep GP), instead of the Bayesian neural networks [1]. However, [1] is not a multi-fidelity HPO method and has no acquisition function. Simply put, [1] is not a multi-fidelity HPO baseline.
>
> In addition, the surrogate model of [1] has no embedding of the learning curve pattern, but only of the budget indicator (i.e. following our formalism, [1] uses only j, not Y). Furthermore, our surrogate is trained for a short-term forecast loss (i.e. predict the performance of the next budget), while [1] estimates the full learning curve until convergence. We believe there are major differences in terms of novelty between our method and the suggested combination of [1] and [2].
>
> **On our multi-fidelity acquisition function**
>
> A. To explain the motivation on why we need a multi-fidelity acquisition consider a toy example with 3 configurations, i.e. consider x := the drop-out rate of a network:
>
> x1=0.3 has been evaluated for 5 epochs with validation accuracy Y_1={0.4, 0.5, 0.6, 0.65, 0.7}
>
> x2=0.2 has been evaluated for 2 epochs with validation accuracy Y_2={0.5, 0.6}
>
> x3=0.1 has been evaluated for 2 epochs with validation accuracy Y_3={0.6, 0.63}
>
> B. Furthermore, assume that our surrogate estimates the next validation accuracy
>
> for x1 to be 0.75
>
> for x2 to be 0.7
>
> for x3 to be 0.65
>
> C. The question we would like to solve is:
>
> Based on the past epochs’ validation performances and the next estimated validation performance, which of the configurations x1, x2, x3 should the acquisition continue training further, as the most promising one?
>
> Let us consider the standard Expected Improvement:
> The best score (incumbent) among the evaluated configurations is 0.7 from x1.
> Therefore, the EI of the surrogate estimations from point B above are:
>
> for x1 we get max(0.75-0.7, 0) = 0.05
>
> for x2 we get max(0.7-0.7, 0) = 0
>
> for x3 we get max(0.65-0.7, 0) = 0
>
> Based on a standard EI then we should continue training the configuration x1, which is an expected phenomenon because typically configurations that have had more epochs will have a higher EI.
>
> As a contrast, consider our multi-fidelity Expected Improvement:
> The best score (incumbent) for x1 is 0.7, while for x1 and x2 is 0.6. The difference is that the best score is not the global best, but the best observed until the respective epoch for which the configuration was last trained. Therefore:
>
> for x1 we get max(0.75-0.7, 0) = 0.05
>
> for x2 we get max(0.7-0.6, 0) = 0.1
>
> for x3 we get max(0.65-0.6, 0) = 0.05
>
> According to our novel acquisition, the best configuration to try is x2, not x1. That is intuitionally clear, as the increase of its validation accuracy is steeper {0.5, 0.6} and next estimated is 0.7.
>
> In our experiments the standard EI performs pretty-much similar to the toy example, it simply keeps training the configuration that has had more epochs so far. A standard EI performs in our experiments only marginally better than a random search.
>
> We hope the motivation of adapting the acquisition function to consider the fidelity/budget-based incumbent is clear.
>
> Please feel free to let us know if any questions or concerns still persist, that would prevent you from raising your score toward acceptance.
>
> [1] Learning Curve Prediction with Bayesian Neural Networks, ICLR’17.
>
> [2] https://proceedings.mlr.press/v51/wilson16.html

---

### Official Review · Reviewer_j16y · 2021-11-04

**Correctness:** 3
**Technical Novelty And Significance:** 3
**Empirical Novelty And Significance:** 3
**Recommendation:** 6
**Confidence:** 3

**Main Review:**

### Post rebuttal

Given the detailed rebuttals by the authors, the updated baselines, I'm confident about increasing my rating for this paper. It might aid their arguments, if the authors were to move some sections of the appendix to the main paper.

---------------------------------------------------------------------------------------------------------------------------------------
## Pros:

The paper is quite clear (see the subsequent comments), and easy to follow. The method proposed is simple, and is an intuitive extension to the methods in the literature. The paper is also well placed in the context of previous methods. The experiments section is quite strong in the experiments and baselines covered.


## Cons:


1. The authors start with the motivation that the rank correlation of performance at various budgets is poor. However, this is seemingly contradicted in Fig 7 left where the exclusion of the learning curve in HPO doesn’t worsen the performance much over all the datasets. The authors show experiments where the inclusion of LC leads to better results for some datasets. The authors should either provide references for the poor correlation, or show statistics of how intermediate performance is a poor predictor of final rank.

2. On training the deep convolutional kernel: The neural net used is a MLP with 128 and 256 hidden units. This is quite a large network. How do the authors reliably train this network with only a few (xi, j, Yi, j-1) tuples? In the initial phases, how reliable are the networks predictions to terminate a run? How are the hyperparameters of this training chosen? The authors should give details, and report ablations on network size, architecture, training parameters (lr, batch size etc). In the absence of these, it is hard to judge the merits of the proposed \phi, as I do not understand how these specifics were arrived at. Also, the authors should report how the additional time of training the deep kernel changes the wall clock time measurements.

3. Evaluations: The use of Epochs and Steps to describe the x-axis in various plots is a little confusing. Are these the same? Also, it is ideal that the authors include the true performance (say test accuracy on CIFAR/ImageNet exps), and the true wall-clock times somewhere in the paper, in addition to the regret plots presented. Also the proposed method’s rank fluctuates quite a bit in the initial steps in Fig 3 and 5; can the authors comment on this?
4. Minor:
“Reversing the training update steps” in the Intro para 1 makes it sound like undoing the update steps.
The authors might consider rewording Para 3 of Motivation to aid readability, as it took me a few reads to grasp the point.
The authors say “gradient descent and Adam” on Page 5 last para, and “gradient ascent and Adam” in A.4.

5. Additional comments: These are general comments the authors might consider discussing.
Do the authors find that the trained deep conv kernel is transferable across tasks? This might have interesting implications, if it can be.
The authors write at the end of Section 6 “the additional use of an explicit learning curve representation might not lead to a strong improvement in every scenario”. While experimental evidence has been provided, can the authors describe what factors determine if incorporating LC dynamics leads to better HPO?


**Summary Of The Paper:**

The paper presents an extension to the commonly used Gaussian Process based HPO by incorporating the learning curve dynamics to decide the next HP configuration to be tried out. For this, the authors propose the use of a kernel that encodes the previous HP iterates using a neural network. The method is shown to reach lower regret values for the same computational budget compared to the baselines considered.


**Summary Of The Review:**

The presented work is interesting, barring a few points commented on above. The motivation for this work needs further clarification from the authors. The experimental evidence of the efficacy of the method is strong. However, the paper in its current form misses some important details, and ablations. If the authors can address these points adequately in their rebuttal, I’d be quite happy to raise my score.

---

> ### Author Response · Authors · 2021-11-20
> **Rebuttal to Reviewer j16y**
>
> Thank you very much for your comments.
>
> **On the poor correlation of configurations:**
>
> We provide empirical evidence on the phenomenon of poor correlations with the aid of the new experiment in Appendix D, illustrated with Figure 16 in the updated manuscript draft. The new experimental evidence highlights the poor correlation phenomenon and demonstrates that DyHPO can handle it efficiently. Because multiple reviewers raised the point we answered this concern through a separate comment at https://openreview.net/forum?id=aBAgwom5pTn&noteId=lXv7bQwUdRi
>
> **On training the deep GP:**
>
> We emphasize that the myth that "deep Gaussian Processes cannot be trained from a limited number of observations" is contradicted by the empirical evidence of recent papers [1,2]. Deep GP models achieve state-of-the-art results on black-box HPO, even under a small data regime (e.g. check the performance of Deep Kernel GP in [2]). Therefore, the empirical success of our deep GP surrogate on a small data regime is aligned with the recent findings of the community. You might find the detailed answer to reviewer RUAE beneficial on the matter https://openreview.net/forum?id=aBAgwom5pTn&noteId=oUMiJV8NyjC
>
> **On the architecture choice:**
>
> Our choice of architecture is a direct evolution of the prior deep GP kernels from [1,2], which are extended for the grey-box HPO setup by incorporating the budget and learning curve information. Please find a detailed motivation of our architecture as an answer to the comments of Reviewer cMi3 at https://openreview.net/forum?id=aBAgwom5pTn&noteId=kJIzDwaP9rQ
>
> **On the additional overhead time of fitting the deep GP:**
>
> Refitting a deep GP adds an overhead to the overall training time. In order to show that the overhead is not significant, we have re-run the LCBench experiments and presented the new aggregated wall-clock results in Figure 1, as well as the detailed wall-clock results of each dataset in Figure 12 and 13. Furthermore, the wall-clock results are shown for the datasets of the NASBench201 tasks. Regarding the TaskSet problems, the benchmark does not offer the runtime for an evaluation, therefore we use the number of steps as a runtime proxy, as provided in the benchmark.
>
> Regarding the true performance, please notice that the regret is a direct proxy of the classification accuracy because it is defined as the (best accuracy - achieved accuracy). In that manner, a regret of 0 means the best-discovered configuration by an HPO method matches the best accuracy of the optimal configuration on a dataset (based on the configurations evaluated in the benchmarks).
>
> [1] FSBO https://openreview.net/forum?id=bJxgv5C3sYc
> [2] HPO-B https://openreview.net/forum?id=O24OhmqpJtP

---

> > ### Comment · Reviewer_j16y · 2021-11-21
> > **RE: Authors' rebuttal**
> >
> > Thank you for the additional details.
> >
> > In Fig 14, 16, for the Taskset plots you have a noticeable change in the plots in the last few epochs. Can the authors shed some light on why this happens?

---

> > > ### Author Response · Authors · 2021-11-23
> > > **Second rebuttal to Reviewer j16y**
> > >
> > > We investigated the behavior you reported on the TaskSet benchmark and notice that there were two reasons for the jump in some metrics of Figures 14, 16 around epochs 40-48.
> > >
> > > 1. Divergence of learning curves from the benchmark
> > >
> > > TaskSet is a benchmark where various Adam Optimizer hyperparameters configurations (learning rate, momentum terms, etc.) are evaluated on different NLP datasets for an LSTM-based model.  In some of those configurations, the validation loss suddenly increases after ca. 40 epochs as a result of divergence due to the specific combination of those configurations' Adam hyperparameters. To check the phenomenon, we share an illustration of the learning curves for the TaskSet configurations selected by DyHPO on the dataset FixedTextRNNClassification_imdb_patch32_LSTM128_bs128:
> > >
> > > https://github.com/ConferencePaperSubmission/DYHPO/blob/main/plots/taskset_curves.pdf
> > >
> > > DyHPO has selected these configurations because of their initially low validation loss. However, as the loss values of the selected configurations diverge, DyHPO's metric on these configurations' quality also drop. Please note that the divergence is a characteristic of the dataset's learning curves, not a specific behavior introduced by our HPO method.
> > >
> > > In contrast, since HB and BOHB do not have dynamic budgets, they are not influenced by the fluctuations around epochs 40-48, because the bracket budgets are 2, 6, 17, 50. Therefore, they jump over this region of divergence at epochs 40-48.  In contrast, DyHPO is influenced by these diverging learning curves because it has an epoch-wise dynamic budget.
> > >
> > > However, we stress that DyHPO still is by far superior to the baselines in all the quality metrics of Figures 14, 15, 16, despite the few diverging learning curves.
> > >
> > > 2. A discrepancy in the code that plots these Figures
> > >
> > > Furthermore, while inspecting the results, we found a bug in the plotting code for Figures 14, 15, 16 for HB and BOHB. For some brackets, the returned bracket budget was 0.999999 instead of 1.0, which caused a precision issue when casting to int. We fixed that discrepancy by rounding the bracket budgets.
> > >
> > > For the sake of double-checking, we re-ran all the paper experiments again for HB, BOHB; and recreated all the plots, including Figures 14-16.
> > >
> > > The updated manuscript is uploaded to OpenReview.
> > >
> > > The updated results do not show any noteworthy difference in the relative performance between the baselines and our method DyHPO, which still maintains a statistically-significant gain against all the baselines.
> > >
> > > ----------------
> > >
> > > Your comments and review were quite helpful. Due to your diligence, we improved and double-checked the quality of the experiments.
> > >
> > > Thanks a lot for assisting us in raising the quality bar of this paper even further.
> > >
> > > Feel free to raise any additional concerns that are still unanswered.

---

> > > > ### Comment · Reviewer_j16y · 2021-11-25
> > > > **Re: More clarifications**
> > > >
> > > > 1. Your argument about the upward tick in 16(b) (TaskSet) plot is still unclear to me. If a configuration diverges say 40th epoch, wouldn't your kernel decide not to train any longer?
> > > >
> > > > 2. "The precision at an epoch i is defined as the number of top 1% candidates trained for at least i epochs
> > > > divided by the number of all candidates trained for at least i epochs" --  can you clarify this? Top 1% candidates at the end of the full budget? If this is the case, why does 14(ii) have a downward tick? Those late diverging configs don't make it into the final 1%.
> > > >
> > > > Thank you for double checking your experiments, and rectifying the bugs you found.

---

> > > > > ### Author Response · Authors · 2021-11-26
> > > > > **More clarifications**
> > > > >
> > > > > Thanks for pointing out this detail and sorry for the confusion created by the perhaps vaguely-formulated description in the previous answer at point "1. Divergence of learning curves from the benchmark". We will try to describe it more clearly now.
> > > > >
> > > > > A portion of configuration learning curves in TaskSet exhibit a low validation loss up to a certain epoch, and then diverge (typically after ca. 40 epochs).  Our method DyHPO selects such configurations because of their initially low validation loss and then stops training them further. Please consider this new illustration of DyHPO on the learning curves we shared in the previous post:
> > > > >
> > > > > https://github.com/ConferencePaperSubmission/DYHPO/blob/main/plots/taskset_curves_dyhpo.pdf
> > > > >
> > > > > Solid parts of the lines indicate the epochs where a configuration was selected by DyHPO, and the dashed line is the remaining learning curve of the configuration from the benchmark. DyHPO does not select the configurations for the epochs indicated with the dashed lines.
> > > > >
> > > > > As a result, the precision drops at later epochs because we define precision as the fraction of top 1% configurations at epoch B, which are still "active" until epoch B. Good top-1% configurations that are stopped before epoch B, are not counted in the precision at epoch B. Therefore, the precision drops at epoch B, because a smaller fraction of the top 1% configurations is still active.
> > > > >
> > > > > We believe early-stopping a configuration is a strength of our method, although it creates a visual artifact due to the definition of the precision metric that includes only active (not early-stopped) configurations.
> > > > >
> > > > > In contrast, as we clarified in the previous response, baselines such as HB, BOHB have predefined budgets at epochs 2, 6, 17, 50. Therefore, you do not see a fluctuation of their precision between epochs 40-50.

---

> ### Author Response · Authors · 2021-12-02
> **Thanks for your suggestion**
>
> Thanks a lot for increasing your score after considering the additional experiments and analyses of our rebuttal. Following your recommendations, we will incorporate the contents of the analyses on Appendix D, Figs 14,15,16, into the main manuscript for the camera ready version.

---

### Author Response · Authors · 2021-11-20
**On poor correlation and the effectiveness of our method**

Reviewers j16y, oS3h, RUAE raised criticism regarding our claims on the poor correlation phenomenon. We are summarizing the concern as two questions:

- Does a poor correlation phenomenon among the performances of configurations at different budgets exist?
- Can DyHPO handle this poor correlation phenomenon?

We believe the criticism is well-grounded, and the reviewers are right in demanding empirical evidence on the matter.

To address the criticism, we are presenting new results in Appendices C and D of the updated manuscript.

Appendix D demonstrates the existence of the poor correlation phenomenon. The metric we introduce is the fraction of "top" performing configurations at a particular budget, that were not "top" performers at one of the lower budgets (details in Appendix D). Furthermore, we show that our method DyHPO handles the phenomenon better than baselines.

In addition, we provide further analysis of the quality of our method in Appendix C, where we compare rivaling methods in terms of the overall quality of the selected hyperparameter configurations, both in terms of the precision and cumulative mean regret.

This new evidence demonstrates that DyHPO selects more top configurations than rivaling methods (Appendix C), even if they perform poorly at lower budgets/fidelities (Appendix D).

---

### Author Response · Authors · 2021-11-30
**Extended results with new baselines**

We would like to present new results that incorporate two additional baselines:

- MF-DNN [1] requested by reviewer cMi3.
- ASHA [2] requested by reviewer RUAE.

The implementation details of the new baselines MF-DNN and ASHA are offered in https://github.com/ConferencePaperSubmission/DYHPO/blob/main/plots/info.md

Please find below the comparison of all the baselines including MF-DNN and ASHA in terms of the wall-clock time and the number of epochs/steps.

-The epochs/step plots show the regret/quality of the discovered hyperparameter configurations as a function of the number of optimization epochs/steps, without taking into account the additional overhead.
- The wallclock plots include the time that a method needs to recommend a configuration (overhead of the method), plus the wallclock time needed to evaluate the recommended configuration.


i. The epochs/steps results on LCBench and TaskSet:

https://github.com/ConferencePaperSubmission/DYHPO/blob/main/plots/lcbench_epochs.pdf

https://github.com/ConferencePaperSubmission/DYHPO/blob/main/plots/taskset_steps.pdf

Note: A step in the TaskSet meta-dataset is different from an epoch in LCBench. The authors of the TaskSet [3] benchmark evaluate hyperparameter configurations after every 200 iterations (details in Appendix A1).

ii. The epochs results on NASBench201 datasets:

https://github.com/ConferencePaperSubmission/DYHPO/blob/main/plots/cifar10_epochs.pdf

https://github.com/ConferencePaperSubmission/DYHPO/blob/main/plots/cifar100_epochs.pdf

https://github.com/ConferencePaperSubmission/DYHPO/blob/main/plots/ImageNet16-120_epochs.pdf

iii. The wallclock results on LCBench:

https://github.com/ConferencePaperSubmission/DYHPO/blob/main/plots/lcbench_time.pdf

iv. The wallclock results on NASBench201 datasets:

https://github.com/ConferencePaperSubmission/DYHPO/blob/main/plots/cifar10_time.pdf

https://github.com/ConferencePaperSubmission/DYHPO/blob/main/plots/cifar100_time.pdf

https://github.com/ConferencePaperSubmission/DYHPO/blob/main/plots/ImageNet16-120_time.pdf

Note: There are no wallclock results on TaskSet because the benchmark offers only the evaluations of hyperparameter configurations, but does not offer the runtimes on how long evaluations took to complete.

Outcome:

The results of the extended experiments with the new baselines indicate that our method DyHPO is still superior to all the baselines. The new baselines MF-DNN and ASHA are significantly less qualitative compared to our method, therefore, do not change the pitch of the paper.

[1] "Multi-Fidelity Bayesian Optimization via Deep Neural Networks" NeurISP2020

[2] ASHA: https://arxiv.org/abs/1810.05934

[3] TaskSet: https://arxiv.org/abs/2002.11887

---

### Author Response · Authors · 2021-11-30
**Summary of the Rebuttal**


i. New baselines: MF-DNN (asked by reviewer cMi3) and ASHA (asked by reviewer RUAE), which we outperform significantly.

ii. Wall-clock results (asked by reviewers j16y, OS3h, RUAE), therefore, demonstrating that the overhead time of fitting the deep GP surrogate of our method does not play a significant role. Our method outperforms all baselines in terms of quality vs. wallclock time, too.

(The evidence for points i. and ii. is accessible here https://openreview.net/forum?id=aBAgwom5pTn&noteId=MlhgfpdLL9b)

iii. Experiments on handling the poor correlation phenomenon (asked by reviewers j16y, oS3h, RUAE) demonstrate the existence of the poor correlation across fidelities/budgets, and that our method DyHPO handles it significantly more efficiently than the baselines (Figures 14-16 of the uploaded manuscript).

iv. In addition, we empirically show that our paper does make correct claims, which challenges the initially low correctness scores by all the reviewers (2,3,1,2,3). The provided analyses offer clear evidence on the existence of the poor correlation performance and our method's superiority in handling it, demonstrated in a large-scale experimental setup with multiple state-of-the-art baselines.

Therefore, we believe the reviewers' concerns (correctness claims, missing baselines, analyses) are answered, and we hope the new reviewing scores will reflect the evidence of the rebuttal.

---

### Decision · Program_Chairs · 2022-01-20

**Decision:**

Reject

**Comment:**

This paper presents a new method for performing Bayesian optimization for hyperparameter tuning that uses learning curve trajectories to reason about how long to train a model for (thus "grey box" optimization) and whether to continue training a model.  The reviewers seem to find the paper clear, well-motivated and the presented methodology sensible.  However, the reviews were quite mixed and leaning towards reject with 3, 6, 5, 3, 6.  A challenge for the authors is that there is already significant related literature on the subject of multi-fidelity optimization and even specific formulations for hyperparameter optimization that reason about learning curves.  A common criticism raised by the reviewers is that while there are extensive experiments, they don't seem to be the right choice of experiments to help understand the advantages of this method (e.g. epochs instead of wall-clock on the x-axis, choice of baselines, demonstration that early results are used to forecast later success, etc.).  Unfortunately, because there is significant related literature, the bar is raised somewhat in terms of empirical evidence (although theoretical evidence of the performance of this method would also help).  It seems clear that some of the reviewers are not convinced by the experiments that were presented.  Thus the recommendation is to reject the paper but encourage the authors to submit to a future venue.  It looks like the authors have gone a long way to address these concerns in their author responses.  Incorporating these new results and the reviewer feedback would go a long way to improving the paper for a future submission.